# Model Sparsity Can Simplify Machine Unlearning

**Jinghan Jia**[1,⋆]    **Jiancheng Liu**[1,⋆]    **Parikshit Ram**[2]    **Yuguang Yao**[1]    **Gaowen Liu**[3]

**Yang Liu**[4,5]                **Pranay Sharma**[6]                **Sijia Liu**[1,2]

[1]Michigan State University, [2]IBM Research, [3]Cisco Research,
[4]University of California, Santa Cruz, [5]ByteDance Research, [6]Carnegie Mellon University
[⋆]Equal contribution

## Abstract

In response to recent data regulation requirements, machine unlearning (MU) has emerged as a critical process to remove the influence of specific examples from a given model. Although exact unlearning can be achieved through complete model retraining using the remaining dataset, the associated computational costs have driven the development of efficient, approximate unlearning techniques. Moving beyond data-centric MU approaches, our study introduces a novel model-based perspective: model sparsification via weight pruning, which is capable of reducing the gap between exact unlearning and approximate unlearning. We show in both theory and practice that model sparsity can boost the multi-criteria unlearning performance of an approximate unlearner, closing the approximation gap, while continuing to be efficient. This leads to a new MU paradigm, termed prune first, then unlearn, which infuses a sparse model prior into the unlearning process. Building on this insight, we also develop a sparsity-aware unlearning method that utilizes sparsity regularization to enhance the training process of approximate unlearning. Extensive experiments show that our proposals consistently benefit MU in various unlearning scenarios. A notable highlight is the 77% unlearning efficacy gain of fine-tuning (one of the simplest unlearning methods) when using sparsity-aware unlearning. Furthermore, we demonstrate the practical impact of our proposed MU methods in addressing other machine learning challenges, such as defending against backdoor attacks and enhancing transfer learning. Codes are available at https://github.com/OPTML-Group/Unlearn-Sparse.

## 1   Introduction

Machine unlearning (**MU**) initiates a reverse learning process to scrub the influence of data points from a trained machine learning (**ML**) model. It was introduced to avoid information leakage about private data upon completion of training [1–3], particularly in compliance with legislation like 'the right to be forgotten' [4] in General Data Protection Regulation (GDPR) [5]. The *direct but optimal* unlearning approach is *exact unlearning* to *retrain* ML models from scratch using the remaining training set, after removing the data points to be scrubbed. Although retraining yields the *ground-truth* unlearning strategy, it is the most computationally intensive one. Therefore, the development of *approximate but fast* unlearning methods has become a major focus in research [6–10].

Despite the computational benefits of approximate unlearning, it often lacks a strong guarantee on the effectiveness of unlearning, resulting in a performance gap with exact unlearning [11]. In particular, we encounter two main challenges. *First*, the performance of approximate unlearning can heavily rely on the configuration of algorithmic parameters. For example, the Fisher forgetting method [12] needs

37th Conference on Neural Information Processing Systems (NeurIPS 2023).

to carefully tune the Fisher information regularization parameter in each data-model setup. *Second*, the effectiveness of an approximate scheme can vary significantly across the different unlearning evaluation criteria, and their trade-offs are not well understood. For example, high 'efficacy' (ability to protect the privacy of the scrubbed data) *neither* implies *nor* precludes high 'fidelity' (accuracy on the remaining dataset) [9]. This raises our **driving question (Q)** below:

> **(Q)** *Is there a theoretically-grounded and broadly-applicable method to improve approximate unlearning across different unlearning criteria?*

To address **(Q)**, we advance MU through a fresh and novel viewpoint: **model sparsification**. *Our key finding* is that model sparsity (achieved by weight pruning) can significantly reduce the gap between approximate unlearning and exact unlearning; see **Fig. 1** for the schematic overview of our proposal and highlighted empirical performance.

Model sparsification (or weight pruning) has been extensively studied in the literature [13–19], focusing on the interrelation between model compression and generalization. For example, the notable lottery ticket hypothesis (**LTH**) [15] demonstrated the existence of a sparse subnetwork (the so-called 'winning ticket') that matches or even exceeds the test accuracy of the original dense model. In addition to generalization, the impact of pruning has also been investigated on model robustness [20–22], fairness [23, 24], interpretability [25, 26], loss landscape [16, 26], and privacy [27, 28]. In particular, the privacy gains from pruning [27, 28] imply connections between data influence and model sparsification.

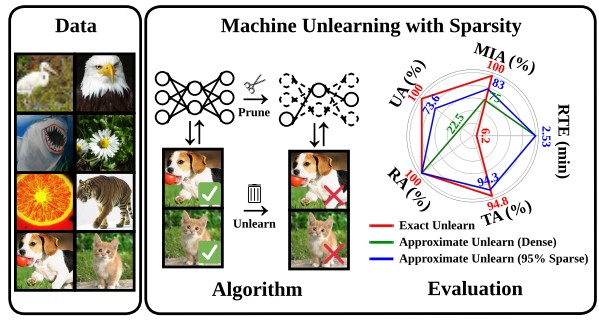

Figure 1: Schematic overview of our proposal on model sparsity-driven MU. Evaluation at-a-glance shows the performance of three unlearning methods (retraining-based exact unlearning, finetuning-based approximate unlearning [12], and proposed unlearning on 95%-sparse model) under five metrics: unlearning accuracy (UA), membership inference attack (MIA)-based unlearning efficacy, accuracy on remaining data (RA), testing accuracy (TA), and run-time efficiency (RTE); see summary in **Tab. 1**. The unlearning scenario is given by class-wise forgetting, where data points of a single class are scrubbed. Each metric is normalized to [0, 1] based on the best result across unlearning methods for ease of visualization. Results indicate that model sparsity reduces the gap between exact and approximate MU without loss in efficiency.

More recently, a few works [29, 30] attempted to draw insights from pruning for unlearning. In Wang et al. [29], removing channels of a deep neural network (**DNN**) showed an unlearning benefit in federated learning. And in Ye et al. [30], filter pruning was introduced in lifelong learning to detect "pruning identified exemplars" [31] that are easy to forget. **However**, different from the above literature that customized model pruning for a specific unlearning application, our work systematically and comprehensively explores and exploits the foundational connections between unlearning and pruning. We summarize our **contributions** below.

● First, we provide a holistic understanding of MU across the full training/evaluation stack.

● Second, we draw a tight connection between MU and model pruning and show in theory and practice that model sparsity helps close the gap between approximate unlearning and exact unlearning.

● Third, we develop a new MU paradigm termed 'prune first, then unlearn', and investigate the influence of pruning methods in the performance of unlearning. Additionally, we develop a novel 'sparsity-aware unlearning' framework that leverages a soft sparsity regularization scheme to enhance the approximate unlearning process.

● Finally, we perform extensive experiments across diverse datasets, models, and unlearning scenarios. Our findings consistently highlight the crucial role of model sparsity in enhancing MU.

## 2 Revisiting Machine Unlearning and Evaluation

**Problem setup.** MU aims to remove (or scrub) the influence of some targeted training data on a trained ML model [1, 2]. Let $\mathcal{D} = \{\mathbf{z}_i\}_{i=1}^N$ be a (training) dataset of $N$ data points, with label

information encoded for supervised learning. $\mathcal{D}_f \subseteq \mathcal{D}$ represents a subset whose influence we want to scrub, termed the **forgetting dataset**. Accordingly, the complement of $\mathcal{D}_f$ is the **remaining dataset**, *i.e.*, $\mathcal{D}_r = \mathcal{D} \setminus \mathcal{D}_f$. We denote by $\boldsymbol{\theta}$ the model parameters, and $\boldsymbol{\theta}_o$ the **original model** trained on the entire training set $\mathcal{D}$ using *e.g.*, empirical risk minimization (ERM). Similarly, we denote by $\boldsymbol{\theta}_u$ an **unlearned model**, obtained by a scrubbing algorithm, after removing the influence of $\mathcal{D}_f$ from the trained model $\boldsymbol{\theta}_o$. The **problem of MU** is to find an accurate and efficient scrubbing mechanism to generate $\boldsymbol{\theta}_u$ from $\boldsymbol{\theta}_o$. In existing studies [2, 7, 12], the choice of the forgetting dataset $\mathcal{D}_f$ specifies different unlearning scenarios. There exist two main categories. First, *class-wise forgetting* [7, 12] refers to unlearning $\mathcal{D}_f$ consisting of training data points of an entire class. Second, *random data forgetting* corresponds to unlearning $\mathcal{D}_f$ given by a subset of random data drawn from all classes.

**Exact and approximate MU methods.** The *exact unlearning* method refers to *retraining* the model parameters from *scratch* over the remaining dataset $\mathcal{D}_r$. Although retraining from scratch (that we term **Retrain**) is optimal for MU, it entails a large computational overhead, particularly for DNN training. This problem is alleviated by *approximate unlearning*, an easy-to-compute proxy for Retrain, which has received growing attention. Yet, the boosted computation efficiency comes at the cost of MU's efficacy. We next review some commonly-used approximate unlearning methods that we improve in the sequel by leveraging sparsity; see a summary in **Tab. 1**.

♦ *Fine-tuning (FT)* [6, 12]: Different from Retrain, FT fine-tunes the pre-trained model $\boldsymbol{\theta}_o$ on $\mathcal{D}_r$ using a few training epochs to obtain $\boldsymbol{\theta}_u$. The rationale is that fine-tuning on $\mathcal{D}_r$ initiates the catastrophic forgetting in the model over $\mathcal{D}_f$ as is common in continual learning [33].

♦ *Gradient ascent (GA)* [7, 8]: GA reverses the model training on $\mathcal{D}_f$ by adding the corresponding gradients back to $\boldsymbol{\theta}_o$, *i.e.*, moving $\boldsymbol{\theta}_o$ in the direction of increasing loss for data points to be scrubbed.

Table 1: Summary of approximate unlearning methods considered in this work. The marker '✓' denotes the metric used in previous research. The number in RTE is the run-time cost reduction compared to the cost of Retrain, based on our empirical studies in Sec. 5 on (CIFAR-10, ResNet-18). Note that GA seems better than ours in terms of RTE, but it is less effective in unlearning.

| Unlearning Methods | Evaluation metrics | | | | | Representative work |
|---|---|---|---|---|---|---|
| | UA | MIA-Efficacy | RA | TA | RTE | |
| FT | ✓ | | ✓ | ✓ | 0.06× | [6, 12] |
| GA | ✓ | ✓ | ✓ | ✓ | 0.02× | [7, 8] |
| FF | ✓ | | ✓ | ✓ | 0.9 × | [9, 12] |
| IU | ✓ | | | ✓ | 0.08× | [10, 32] |
| Ours | ✓ | ✓ | ✓ | ✓ | 0.07× | This work |

♦ *Fisher forgetting (FF)* [9, 12]: FF adopts an additive Gaussian noise to 'perturb' $\boldsymbol{\theta}_o$ towards exact unlearning. Here the Gaussian distribution has zero mean and covariance determined by the 4th root of Fisher Information matrix with respect to (w.r.t.) $\boldsymbol{\theta}_o$ on $\mathcal{D}_r$. We note that the computation of the Fisher Information matrix exhibits lower parallel efficiency in contrast to other unlearning methods, resulting in higher computational time when executed on GPUs; see Golatkar et al. [12] for implementation details.

♦ *Influence unlearning (IU)* [10, 32]: IU leverages the influence function approach [34] to characterize the change in $\boldsymbol{\theta}_o$ if a training point is removed from the training loss. IU estimates the change in model parameters from $\boldsymbol{\theta}_o$ to $\boldsymbol{\theta}_u$, *i.e.*, $\boldsymbol{\theta}_u - \boldsymbol{\theta}_o$. IU also relates to an important line of research in MU, known as $\epsilon$-$\delta$ forgetting [29, 35, 36]. However, it typically requires additional model and training assumptions [35].

We next take a step further to revisit the IU method and re-derive its formula (**Prop. 1**), with the aim of enhancing the effectiveness of existing solutions proposed in the previous research.

**Proposition 1** *Given the weighted ERM training* $\boldsymbol{\theta}(\mathbf{w}) = \arg\min_{\boldsymbol{\theta}} L(\mathbf{w}, \boldsymbol{\theta})$ *where* $L(\mathbf{w}, \boldsymbol{\theta}) = \sum_{i=1}^{N} [w_i \ell_i(\boldsymbol{\theta}, \mathbf{z}_i)]$, $w_i \in [0, 1]$ *is the influence weight associated with the data point* $\mathbf{z}_i$ *and* $\mathbf{1}^T \mathbf{w} = 1$, *the model update from* $\boldsymbol{\theta}_o$ *to* $\boldsymbol{\theta}(\mathbf{w})$ *yields*

$$\Delta(\mathbf{w}) := \boldsymbol{\theta}(\mathbf{w}) - \boldsymbol{\theta}_o \approx \mathbf{H}^{-1} \nabla_{\boldsymbol{\theta}} L(\mathbf{1}/N - \mathbf{w}, \boldsymbol{\theta}_o), \tag{1}$$

*where* $\mathbf{1}$ *is the* $N$-*dimensional vector of all ones,* $\mathbf{w} = \mathbf{1}/N$ *signifies the uniform weights used by ERM,* $\mathbf{H}^{-1}$ *is the inverse of the Hessian* $\nabla_{\boldsymbol{\theta},\boldsymbol{\theta}}^2 L(\mathbf{1}/N, \boldsymbol{\theta}_o)$ *evaluated at* $\boldsymbol{\theta}_o$, *and* $\nabla_{\boldsymbol{\theta}} L$ *is the gradient of* $L$. *When scrubbing* $\mathcal{D}_f$, *the unlearned model is given by* $\boldsymbol{\theta}_u = \boldsymbol{\theta}_o + \Delta(\mathbf{w}_{MU})$. *Here* $\mathbf{w}_{MU} \in [0, 1]^N$ *with entries* $w_{MU,i} = \mathbb{I}_{\mathcal{D}_r}(i)/|\mathcal{D}_r|$ *signifying the data influence weights for MU,* $\mathbb{I}_{\mathcal{D}_r}(i)$ *is the indicator function with value 1 if* $i \in \mathcal{D}_r$ *and 0 otherwise, and* $|\mathcal{D}_r|$ *is the cardinality of* $\mathcal{D}_r$.

**Proof**: We derive (1) using an implicit gradient approach; see Appendix A. □

It is worth noting that we have taken into consideration the weight normalization effect $\mathbf{1}^T \mathbf{w} = 1$ in (1). This is different from existing work like Izzo et al. [10, Sec. 3] using Boolean or unbounded

weights. In practice, we found that IU with weight normalization can improve the unlearning performance. Furthermore, to update the model influence given by (1), one needs to acquire the second-order information in the form of inverse-Hessian gradient product. Yet, the exact computation is prohibitively expensive. To overcome this issue, we use the first-order WoodFisher approximation [37] to estimate the inverse-Hessian gradient product.

**Towards a 'full-stack' MU evaluation.** Existing work has assessed MU performance from different aspects [7, 8, 12]. Yet, a single performance metric may provide a limited view of MU [11]. By carefully reviewing the prior art, we focus on the following empirical metrics (summarized in Tab. 1).

✦ *Unlearning accuracy (UA)*: We define $\mathrm{UA}(\boldsymbol{\theta}_{\mathrm{u}}) = 1 - \mathrm{Acc}_{\mathcal{D}_{\mathrm{f}}}(\boldsymbol{\theta}_{\mathrm{u}})$ to characterize the *efficacy* of MU in the accuracy dimension, where $\mathrm{Acc}_{\mathcal{D}_{\mathrm{f}}}(\boldsymbol{\theta}_{\mathrm{u}})$ is the accuracy of $\boldsymbol{\theta}_{\mathrm{u}}$ on the forgetting dataset $\mathcal{D}_{\mathrm{f}}$ [7, 12]. It is important to note that a more favorable UA for an approximate unlearning method should **reduce its performance disparity with the gold-standard retrained model (Retrain)**; a higher value is not necessarily better. This principle also extends to other evaluation metrics.

✦ *Membership inference attack (MIA) on $\mathcal{D}_{\mathrm{f}}$ (MIA-Efficacy)*: This is another metric to assess the *efficacy* of unlearning. It is achieved by applying the confidence-based MIA predictor [38, 39] to the unlearned model ($\boldsymbol{\theta}_{\mathrm{u}}$) on the forgetting dataset ($\mathcal{D}_{\mathrm{f}}$). The MIA success rate can then indicate how many samples in $\mathcal{D}_{\mathrm{f}}$ can be correctly predicted as forgetting (*i.e.*, non-training) samples of $\boldsymbol{\theta}_{\mathrm{u}}$. A *higher* MIA-Efficacy implies less information about $\mathcal{D}_{\mathrm{f}}$ in $\boldsymbol{\theta}_{\mathrm{u}}$; see Appendix C.3 for more details.

✦ *Remaining accuracy (RA)*: This refers to the accuracy of $\boldsymbol{\theta}_{\mathrm{u}}$ on $\mathcal{D}_{\mathrm{r}}$, which reflects the *fidelity* of MU [9], *i.e.*, training data information should be preserved from $\boldsymbol{\theta}_{\mathrm{o}}$ to $\boldsymbol{\theta}_{\mathrm{u}}$.

✦ *Testing accuracy (TA)*: This measures the *generalization* ability of $\boldsymbol{\theta}_{\mathrm{u}}$ on a testing dataset rather than $\mathcal{D}_{\mathrm{f}}$ and $\mathcal{D}_{\mathrm{r}}$. TA is evaluated on the whole test dataset, except for class-wise forgetting, in which testing data points belonging to the forgetting class are not in the testing scope.

✦ *Run-time efficiency (RTE)*: This measures the computation efficiency of an MU method. For example, if we regard the run-time cost of Retrain as the baseline, the computation acceleration gained by different approximate unlearning methods is summarized in Tab. 1.

## 3    Model Sparsity: A Missing Factor Influencing Machine Unlearning

**Model sparsification via weight pruning.** Model sparsification could not only facilitate a model's training, inference, and deployment but also benefit model's performance. For example, LTH (lottery ticket hypothesis) [15] stated that a trainable sparse sub-model could be identified from the original dense model, with test accuracy on par or even better than the original model. **Fig. 2** shows an example of the pruned model's generalization vs. its sparsity ratio. Here one-shot magnitude pruning (**OMP**) [17] is adopted to obtain sparse models. OMP is computationally the lightest pruning method, which directly prunes the model weights to the target sparsity ratio based on their magnitudes. As we can see, there exists a graceful sparse regime with lossless testing accuracy.

**Gains of MU from sparsity.** We first analyze the impact of model sparsity on MU through a lens of *unrolling stochastic gradient descent* (**SGD**) [8]. The specified SGD method allows us to derive the *unlearning error* (given by the weight difference between the approximately unlearned model and the gold-standard retrained model) when scrubbing a single data point. However, different from Thudi et al. [8], we will infuse the model sparsity into SGD unrolling.

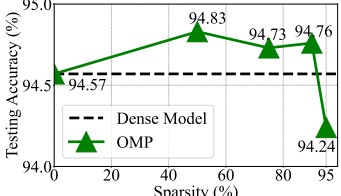

Figure 2: Testing accuracy of OMP-based sparse ResNet-18 vs. the dense model on CIFAR-10.

Let us assume a binary mask $\mathbf{m}$ associated with the model parameters $\boldsymbol{\theta}$, where $m_i = 0$ signifies that the $i$th parameter $\theta_i$ is pruned to zero and $m_i = 1$ represents the unmasked $\theta_i$. This sparse pattern $\mathbf{m}$ could be obtained by a weight pruning method, like OMP. Given $\mathbf{m}$, the **sparse model** is $\mathbf{m} \odot \boldsymbol{\theta}$, where $\odot$ denotes the element-wise multiplication. Thudi *et al.* [8] showed that if GA is adopted to scrub a single data point for the original (dense) model $\boldsymbol{\theta}$ (*i.e.*, $\mathbf{m} = \mathbf{1}$), then the gap between GA and Retrain can be approximately bounded in the weight space. **Prop. 2** extends the existing unlearning error analysis to a sparse model.

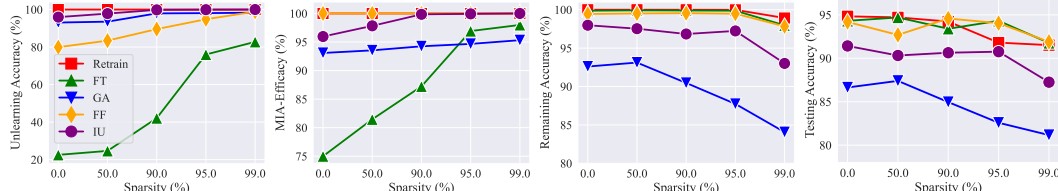

Figure 3: Performance of approximate unlearning (FT, GA, FF, IU) and exact unlearning (Retrain) in efficacy (UA and MIA-Efficacy), fidelity (RA), and generalization (TA) vs. model sparsity (achieved by OMP) in the data-model setup (CIFAR-10, ResNet-18). The unlearning scenario is class-wise forgetting, and the average unlearning performance over 10 classes is reported. We remark that being closer to Retrain performance is better for approximate MU schemes.

**Proposition 2** *Given the model sparse pattern* $\mathbf{m}$ *and the SGD-based training, the unlearning error of GA, denoted by* $e(\mathbf{m})$, *can be characterized by the weight distance between the GA-unlearned model and the gold-standard retrained model. This leads to the error bound*

$$e(\mathbf{m}) = \mathcal{O}(\eta^2 t \|\mathbf{m} \odot (\boldsymbol{\theta}_t - \boldsymbol{\theta}_0)\|_2 \sigma(\mathbf{m})) \tag{2}$$

*where* $\mathcal{O}$ *is the big-O notation,* $\eta$ *is the learning rate,* $t$ *is the number of training iterations,* $(\boldsymbol{\theta}_t - \boldsymbol{\theta}_0)$ *denotes the weight difference at iteration* $t$ *from its initialization* $\boldsymbol{\theta}_0$, *and* $\sigma(\mathbf{m})$ *is the largest singular value* $(\sigma)$ *of the Hessian* $\nabla^2_{\boldsymbol{\theta},\boldsymbol{\theta}} \ell$ *(for a training loss* $\ell$) *among the unmasked parameter dimensions, i.e.,* $\sigma(\mathbf{m}) := \max_j \{\sigma_j(\nabla^2_{\boldsymbol{\theta},\boldsymbol{\theta}} \ell), if\ m_j \neq 0\}$.

***Proof***: *See Appendix B.* □

We next draw some key insights from **Prop. 2**. *First*, it is clear from (2) that the unlearning error reduces as the model sparsity in $\mathbf{m}$ increases. By contrast, the unlearning error derived in Thudi et al. [8] for a dense model (*i.e.*, $\mathbf{m} = \mathbf{1}$) is proportional to the dense model distance $\|\boldsymbol{\theta}_t - \boldsymbol{\theta}_0\|_2$. Thus, model sparsity is beneficial to reducing the gap between (GA-based) approximate and exact unlearning. *Second*, the error bound (2) enables us to relate MU to the spectrum of the Hessian of the loss landscape. The number of active singular values (corresponding to nonzero dimensions in $\mathbf{m}$) decreases when the sparsity grows. However, it is important to note that in a high-sparsity regime, the model's generalization could decrease. Consequently, it is crucial to select the model sparsity to strike a balance between generalization and unlearning performance.

Inspired by Prop. 2, we ask: *Does the above benefit of model sparsification in MU apply to other approximate unlearning methods besides GA?* This drives us to investigate the performance of approximate unlearning across the entire spectrum as depicted in Tab. 1. Therefore, **Fig. 3** shows the unlearning efficacy (UA and MIA-Efficacy), fidelity (RA), and generalization (TA) of different approximate unlearning methods in the sparse model regime. Here class-wise forgetting is considered for MU and OMP is used for weight pruning. As we can see, the efficacy of approximate unlearning is significantly improved as the model sparsity increases, *e.g.*, UA and MIA-Efficacy of using FT over 90% sparsity. By contrast, FT over the dense model (0% sparsity) is the least effective for MU. Also, the efficacy gap between exact unlearning (Retrain) and approximate unlearning reduces on sparse models. Further, through the fidelity and generalization lenses, FT and FF yield the RA and TA performance closest to Retrain, compared to other unlearning methods. In the regime of ultra-high sparsity (99%), the efficacy of unlearning exhibits a tradeoff with RA and TA to some extent.

## 4 Sparsity-Aided Machine Unlearning

Our study in Sec. 3 suggests the new MU paradigm 'prune first, then unlearn', which leverages the fact that (approximate) unlearning on a sparse model yields a smaller unlearning error (Prop. 2) and improves the efficacy of MU (Fig. 3). This promising finding, however, raises some new questions. First, it remains elusive how the choice of a weight pruning method impacts the unlearning performance. Second, it leaves room for developing sparsity-aware MU methods that can directly scrub data influence from a dense model.

**Prune first, then unlearn: Choice of pruning methods.** There exist many ways to find the desired sparse model in addition to OMP. Examples include pruning at random initialization before training [40, 41] and simultaneous pruning-training iterative magnitude pruning (**IMP**) [15]. Thus, the problem of pruning method selection arises for MU. From the viewpoint of MU, the unlearner would

prioritize a pruning method that satisfies the following criteria: ❶ *least dependence* on the forgetting dataset ($\mathcal{D}_f$), ❷ *lossless generalization* when pruning, and ❸ *pruning efficiency*. The rationale behind ❶ is that it is desirable *not* to incorporate information of $\mathcal{D}_f$ when seeking a sparse model prior to unlearning. And the criteria ❷ and ❸ ensure that sparsity cannot hamper TA (testing accuracy) and RTE (run-time efficiency). Based on ❶-❸, we propose to use two pruning methods.

✦ **SynFlow** (synaptic flow pruning) [40]: SynFlow provides a (training-free) pruning method at initialization, even without accessing the dataset. Thus, it is uniquely suited for MU to meet the criterion ❶. And SynFlow is easy to compute and yields a generalization improvement over many other pruning-at-initialization methods; see justifications in [40].

✦ **OMP** (one-shot magnitude pruning) [17]: Different from SynFlow, OMP, which we focused on in Sec. 3, is performed over the original model ($\boldsymbol{\theta}_o$). It may depend on the forgetting dataset ($\mathcal{D}_f$), but has a much weaker dependence compared to IMP-based methods. Moreover, OMP is computationally lightest (*i.e.* best for ❸) and can yield better generalization than SynFlow [18].

Furthermore, it is important to clarify that IMP (iterative magnitude pruning) is *not* suitable for MU, despite being widely used to find the most accurate sparse models (*i.e.*, best for criterion ❷). Compared with the proposed pruning methods, IMP has the largest computation overhead and the strongest correlation with the training dataset (including $\mathcal{D}_f$), thereby deviating from ❶ and ❸. In **Fig. 4**, we show the efficacy of FT-based unlearning on sparse models generated using different pruning methods (SynFlow, OMP, and IMP). As we can see, unlearning on SynFlow or OMP-generated sparse

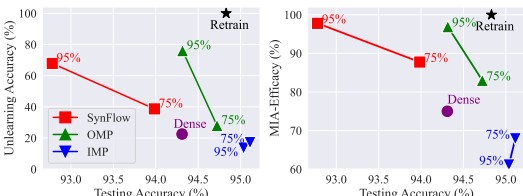

Figure 4: Influence of different pruning methods (Syn-Flow, OMP, and IMP) in unlearning efficacy (UA and MIA-Efficacy) and generalization (TA) on (CIFAR-10, ResNet-18). **Left**: UA vs. TA. **Right**: MIA-Efficacy vs. TA. Each point is a FT-based unlearned dense or sparse model (75% or 95% sparsity), or a retrained dense model.

models yields improved UA and MIA-Efficacy over that on the original dense model and IMP-generated sparse models. This unlearning improvement over the dense model is consistent with Fig. 3. More interestingly, we find that IMP *cannot* benefit the unlearning efficacy, although it leads to the best TA. This is because IMP heavily relies on the training set including forgetting data points, which is revealed by the empirical results – the unlearning metrics get worse for IMP with increasing sparsity. Furthermore, when examining the performance of SynFlow and OMP, we observe that the latter generally outperforms the former, exhibiting results that are closer to those of Retrain. Thus, *OMP is the pruning method we will use by default.*

**Sparsity-aware unlearning.** We next study if pruning and unlearning can be carried out simultaneously, without requiring prior knowledge of model sparsity. Let $L_u(\boldsymbol{\theta}; \boldsymbol{\theta}_o, \mathcal{D}_r)$ denote the unlearning objective function of model parameters $\boldsymbol{\theta}$, given the pre-trained state $\boldsymbol{\theta}_o$, and the remaining training dataset $\mathcal{D}_r$. Inspired by sparsity-inducing optimization [42], we integrate an $\ell_1$ norm-based sparse penalty into $L_u$. This leads to the problem of '$\ell_1$-**sparse MU**':

$$\boldsymbol{\theta}_u = \arg\min_{\boldsymbol{\theta}} L_u(\boldsymbol{\theta}; \boldsymbol{\theta}_o, \mathcal{D}_r) + \gamma\|\boldsymbol{\theta}\|_1, \tag{3}$$

where we specify $L_u$ by the fine-tuning objective, and $\gamma > 0$ is a regularization parameter that controls the penalty level of the $\ell_1$ norm, thereby reducing the magnitudes of 'unimportant' weights.

Table 2: MU performance comparison of using $\ell_1$-sparse MU with different sparsity schedulers of $\gamma$ in (3) and using Retrain. The unlearning scenario is given by random data forgetting (10% data points across all classes) on (ResNet-18, CIFAR-10). A performance gap against Retrain is provided in (●).

| MU | UA | MIA-Efficacy | RA | TA | RTE (min) |
|---|---|---|---|---|---|
| Retrain | 5.41 | 13.12 | 100.00 | 94.42 | 42.15 |
| $\ell_1$-sparse MU + constant $\gamma$ | 6.60 (1.19) | 14.64 (1.52) | 96.51 (3.49) | 87.30 (7.12) | 2.53 |
| $\ell_1$-sparse MU + linear growing $\gamma$ | 3.80 (1.61) | 8.75 (4.37) | 97.13 (2.87) | 90.63 (3.79) | 2.53 |
| $\ell_1$-sparse MU + linear decaying $\gamma$ | 5.35 (0.06) | 12.71 (0.41) | 97.39 (2.61) | 91.26 (3.16) | 2.53 |

In practice, the unlearning performance could be sensitive to the choice of the sparse regularization parameter $\gamma$. To address this limitation, we propose the design of a sparse regularization scheduler. Specifically, we explore three schemes: (1) constant $\gamma$, (2) linearly growing $\gamma$ and (3) linearly decaying $\gamma$; see Sec. 5.1 for detailed implementations. Our empirical evaluation presented in **Tab. 2**

shows that the use of a linearly decreasing $\gamma$ scheduler outperforms other schemes. This scheduler not only minimizes the gap in unlearning efficacy compared to Retrain, but also improves the preservation of RA and TA after unlearning. These findings suggest that it is advantageous to prioritize promoting sparsity during the early stages of unlearning and then gradually shift the focus towards enhancing fine-tuning accuracy on the remaining dataset $\mathcal{D}_r$.

## 5 Experiments

### 5.1 Experiment setups

**Datasets and models.** Unless specified otherwise, our experiments will focus on image classification under CIFAR-10 [43] using ResNet-18 [44]. Yet, experiments on additional datasets (CIFAR-100 [43], SVHN [45], and ImageNet [46]) and an alternative model architecture (VGG-16 [47]) can be found in Appendix C.4. Across all the aforementioned datasets and model architectures, our experiments consistently show that model sparsification can effectively reduce the gap between approximate unlearning and exact unlearning.

**Unlearning and pruning setups.** We focus on two unlearning scenarios mentioned in Sec. 2, *class-wise forgetting* and *random data forgetting* ($10\%$ of the whole training dataset together with 10 random trials). In the '*prune first, then unlearn*' paradigm, we focus on unlearning methods (FT, GA, FF, and IU) shown in Tab. 1 when applying to sparse models. We implement these methods following their official repositories. However, it is worth noting that the implementation of FF in Golatkar et al. [12] modifies the model architecture in class-wise forgetting, *i.e.*, removes the prediction head of the class to be scrubbed. By contrast, other methods keep the model architecture intact during unlearning. Also, we choose OMP as the default pruning method, as justified in Fig. 4. In the '*sparsity-aware unlearning*' paradigm, the sparsity-promoting regularization parameter $\gamma$ in (3) is determined through the line search in the interval $[10^{-5}, 10^{-1}]$, with consideration for the trade-off between testing accuracy and unlearning accuracy. For all schedulers, $\gamma$ is set around to $5 \times 10^{-4}$. The linearly increasing and decaying schedulers are implemented as $\gamma_t = \frac{2t}{T}\gamma$ and $\gamma_t = (2 - \frac{2t}{T})\gamma$ respectively, where $t$ is the epoch index and $T$ is the total number of epochs. We refer readers to Appendix C.2 for more details.

**Evaluation metrics.** We evaluate the unlearning performance following Tab. 1. Recall that UA and MIA-Efficacy depict the *efficacy* of MU, RA reflects the *fidelity* of MU, and TA and RTE characterize the *generalization ability* and the *computation efficiency* of an unlearning method. We implement MIA (membership inference attack) using the prediction confidence-based attack method [38, 39], whose effectiveness has been justified in Song and Mittal [48] compared to other methods. We refer readers to Appendix C.3 for more implementation details. To more precisely gauge the proximity of each approximate MU to Retrain, we introduce a metric termed 'Disparity Average'. This metric quantifies the mean performance gap between each unlearning method and Retrain across all considered metrics. A lower value indicates closer performance to Retrain.

### 5.2 Experiment results

Table 3: Performance overview of various MU methods on dense and 95%-sparse models considering different unlearning scenarios: class-wise forgetting, and random data forgetting. The forgetting data of random data forgetting ratio is $10\%$ of the whole training dataset, the sparse models are obtained using OMP [17], and the unlearning methods and evaluation metrics are summarized in Tab. 1. Class-wise forgetting is conducted class-wise. The performance is reported in the form $a_{\pm b}$, with mean $a$ and standard deviation $b$ computed over 10 independent trials. A performance gap against Retrain is provided in (●). Note that the better performance of approximate unlearning corresponds to the smaller performance gap with the gold-standard retrained model. 'Disparity Ave.' represents the average unlearning gaps across diverse metrics.

| MU | UA DENSE | UA 95% Sparsity | MIA-Efficacy DENSE | MIA-Efficacy 95% Sparsity | RA DENSE | RA 95% Sparsity | TA DENSE | TA 95% Sparsity | Disparity Ave. ↓ DENSE | Disparity Ave. ↓ 95% Sparsity | RTE (min) |
|---|---|---|---|---|---|---|---|---|---|---|---|
| | | | | | Class-wise forgetting | | | | | | |
| Retrain | $100.00_{\pm 0.00}$ | $100.00_{\pm 0.00}$ | $100.00_{\pm 0.00}$ | $100.00_{\pm 0.00}$ | $100.00_{\pm 0.00}$ | $99.99_{\pm 0.01}$ | $94.83_{\pm 0.11}$ | $91.80_{\pm 0.89}$ | 0.00 | 0.00 | 43.23 |
| FT | $22.53_{\pm 8.16}$ (77.47) | $73.64_{\pm 9.46}$ (26.36) | $75.00_{\pm 14.68}$ (25.00) | $83.02_{\pm 16.33}$ (16.98) | $99.87_{\pm 0.04}$ (0.13) | $99.87_{\pm 0.05}$ (0.12) | $94.31_{\pm 0.19}$ (0.52) | $94.32_{\pm 0.12}$ (2.52) | 25.78 | 11.50 | 2.52 |
| GA | $93.08_{\pm 2.29}$ (6.92) | $98.09_{\pm 1.11}$ (1.91) | $94.03_{\pm 3.27}$ (5.97) | $97.74_{\pm 2.24}$ (2.26) | $92.60_{\pm 0.25}$ (7.40) | $87.74_{\pm 0.27}$ (12.25) | $86.64_{\pm 0.28}$ (8.19) | $82.58_{\pm 0.27}$ (9.22) | 7.12 | 6.41 | 0.33 |
| FF | $79.93_{\pm 8.92}$ (20.07) | $94.83_{\pm 4.29}$ (5.17) | $100.00_{\pm 0.00}$ (0.00) | $100.00_{\pm 0.00}$ (0.00) | $99.45_{\pm 0.24}$ (0.55) | $99.48_{\pm 0.33}$ (0.51) | $94.18_{\pm 0.08}$ (0.65) | $94.04_{\pm 0.10}$ (2.24) | 5.32 | 1.98 | 38.91 |
| IU | $87.82_{\pm 2.15}$ (12.18) | $99.47_{\pm 0.15}$ (0.53) | $95.96_{\pm 0.21}$ (4.04) | $99.93_{\pm 0.04}$ (0.07) | $97.98_{\pm 0.21}$ (2.02) | $97.24_{\pm 0.13}$ (2.75) | $91.42_{\pm 0.21}$ (3.41) | $90.76_{\pm 0.18}$ (1.04) | 5.41 | 1.10 | 3.25 |
| | | | | | Random data forgetting | | | | | | |
| Retrain | $5.41_{\pm 0.11}$ | $6.77_{\pm 0.23}$ | $13.12_{\pm 0.14}$ | $14.17_{\pm 0.18}$ | $100.00_{\pm 0.00}$ | $100.00_{\pm 0.00}$ | $94.42_{\pm 0.09}$ | $93.33_{\pm 0.12}$ | 0.00 | 0.00 | 42.15 |
| FT | $6.83_{\pm 0.51}$ (1.42) | $5.97_{\pm 0.57}$ (0.80) | $14.97_{\pm 0.62}$ (1.85) | $13.36_{\pm 0.59}$ (0.81) | $96.61_{\pm 0.25}$ (3.39) | $96.99_{\pm 0.31}$ (3.01) | $90.13_{\pm 0.26}$ (4.29) | $90.29_{\pm 0.31}$ (3.04) | 2.74 | 1.92 | 2.33 |
| GA | $7.54_{\pm 0.29}$ (2.13) | $5.62_{\pm 0.46}$ (1.15) | $10.04_{\pm 0.31}$ (3.08) | $11.76_{\pm 0.52}$ (2.41) | $93.31_{\pm 0.04}$ (6.69) | $95.44_{\pm 0.11}$ (4.56) | $89.28_{\pm 0.07}$ (5.14) | $89.26_{\pm 0.15}$ (4.07) | 4.26 | 3.05 | 0.31 |
| FF | $7.84_{\pm 0.71}$ (2.43) | $8.16_{\pm 0.67}$ (1.39) | $9.52_{\pm 0.43}$ (3.60) | $10.80_{\pm 0.37}$ (3.37) | $92.05_{\pm 0.16}$ (7.95) | $92.29_{\pm 0.24}$ (7.71) | $88.10_{\pm 0.19}$ (6.32) | $87.79_{\pm 0.23}$ (5.54) | 5.08 | 4.50 | 38.24 |
| IU | $2.03_{\pm 0.43}$ (3.38) | $6.51_{\pm 0.52}$ (0.26) | $5.07_{\pm 0.74}$ (8.05) | $11.93_{\pm 0.68}$ (2.24) | $98.26_{\pm 0.29}$ (1.74) | $94.94_{\pm 0.31}$ (5.06) | $91.33_{\pm 0.22}$ (3.09) | $88.74_{\pm 0.42}$ (4.59) | 4.07 | 3.08 | 3.22 |

**Model sparsity improves approximate unlearning.** In **Tab. 3**, we study the impact of model sparsity on the performance of various MU methods in the 'prune first, then unlearn' paradigm. The performance of the exact unlearning method (Retrain) is also provided for comparison. Note that the better performance of approximate unlearning corresponds to the smaller performance gap with the gold-standard retrained model.

*First*, given an approximate unlearning method (FT, GA, FF, or IU), we consistently observe that model sparsity improves UA and MIA-Efficacy (*i.e.*, the efficacy of approximate unlearning) without much performance loss in RA (*i.e.*, fidelity). In particular, the performance gap between each approximate unlearning method and Retrain reduces as the model becomes sparser (see the '95% sparsity' column vs. the 'dense' column). Note that the performance gap against Retrain is highlighted in (·) for each approximate unlearning. We also observe that Retrain on the 95%-sparsity model encounters a 3% TA drop. Yet, from the perspective of approximate unlearning, this drop brings in a more significant improvement in UA and MIA-Efficacy when model sparsity is promoted. Let us take FT (the simplest unlearning method) for class-wise forgetting as an example. As the model sparsity reaches 95%, we obtain 51% UA improvement and 8% MIA-Efficacy improvement. Furthermore, FT and IU on the 95%-sparsity model can better preserve TA compared to other methods. Table 3 further indicates that sparsity reduces average disparity compared to a dense model across various approximate MU methods and unlearning scenarios.

*Second*, existing approximate unlearning methods have different pros and cons. Let us focus on the regime of 95% sparsity. We observe that FT typically yields the best RA and TA, which has a tradeoff with its unlearning efficacy (UA and MIA-Efficacy). Moreover, GA yields the worst RA since it is most loosely connected with the remaining dataset $\mathcal{D}_r$. FF becomes ineffective when scrubbing random data points compared to its class-wise unlearning performance. Furthermore, IU causes a TA drop but yields the smallest gap with exact unlearning across diverse metrics under the 95% model sparsity. In Appendix C.4, we provide additional results on CIFAR-100 and SVHN datasets, as shown in Tab. A3, as well as on the ImageNet dataset, depicted in Tab. A5. Other results pertaining to the VGG-16 architecture are provided in Tab. A4.

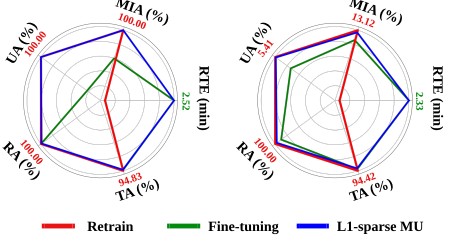

(a) Class-wise forgetting    (b) Random data forgetting

Figure 5: Performance of sparsity-aware unlearning vs. FT and Retrain on class-wise forgetting and random data forgetting under (CIFAR-10, ResNet-18). Each metric is normalized to $[0, 1]$ based on the best result across unlearning methods for ease of visualization, while the actual best value is provided (*e.g.*, 2.52 is the least computation time for class-wise forgetting).

**Effectiveness of sparsity-aware unlearning.** In **Fig. 5**, we showcase the effectiveness of the proposed sparsity-aware unlearning method, *i.e.*, $\ell_1$-sparse MU. For ease of presentation, we focus on the comparison with FT and the optimal Retrain strategy in both class-wise forgetting and random data forgetting scenarios under (CIFAR-10, ResNet-18). As we can see, $\ell_1$-sparse MU outperforms FT in the unlearning efficacy (UA and MIA-Efficacy), and closes the performance gap with Retrain without losing the computation advantage of approximate unlearning. We refer readers to Appendix C.4 and Fig. A2 for further exploration of $\ell_1$-sparse MU on additional datasets.

**Application: MU for Trojan model cleanse.** We next present an application of MU to remove the influence of poisoned backdoor data from a learned model, following the backdoor attack setup [49], where an adversary manipulates a small portion of training data (*a.k.a.* poisoning ratio) by injecting a backdoor trigger (*e.g.*, a small image patch) and modifying data labels towards a targeted incorrect label. The trained model is called *Trojan model*, yielding the backdoor-designated incorrect prediction if the trigger is present at testing. Otherwise, it behaves normally.

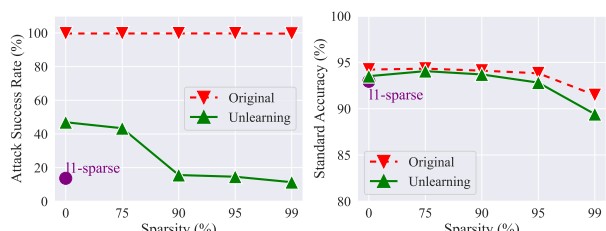

Figure 6: Performance of Trojan model cleanse via proposed unlearning vs. model sparsity, where 'Original' refers to the original Trojan model. **Left**: ASR vs. model sparsity. **Right**: SA vs. model sparsity.

We then regard MU as a defensive method to scrub the harmful influence of poisoned training data in the model's prediction, with a similar motivation as Liu et al. [50]. We evaluate the performance of the unlearned model from two perspectives, backdoor attack success rate (**ASR**) and standard accuracy (**SA**). **Fig. 6** shows ASR and SA of the Trojan model (with poisoning ratio 10%) and its unlearned version using the simplest FT method against model sparsity. Fig. 6 also includes the $\ell_1$-sparse MU to demonstrate its effectiveness on model cleanse. Since it is applied to a dense model (without using hard thresholding to force weight sparsity), it contributes just a single data point at the sparsity level 0%. As we can see, the original Trojan model maintains 100% ASR and a similar SA across different model sparsity levels. By contrast, FT-based unlearning can reduce ASR without inducing much SA loss. Such a defensive advantage becomes more significant when sparsity reaches 90%. Besides, $\ell_1$-sparse MU can also effectively remove the backdoor effect while largely preserving the model's generalization. Thus, our proposed unlearning shows promise in application of backdoor attack defense.

**Application: MU to improve transfer learning.** Further, we utilize the $\ell_1$-sparse MU method to mitigate the impact of harmful data classes of ImageNet on transfer learning. This approach is inspired by Jain et al. [51], which shows that removing specific negatively-influenced ImageNet classes and retraining a source model can enhance its transfer learning accuracy on downstream datasets after finetuning. However, retraining the source model introduces additional computational overhead. MU naturally addresses this limitation and offers a solution.

**Tab. 4** illustrates the transfer learning accuracy of the unlearned or retrained source model (ResNet-18) on ImageNet, with $n$ classes removed. The downstream target datasets used for evaluation are SUN397 [52] and OxfordPets [53]. The employed finetuning approach is linear probing, which finetunes the classification head of the source model on target datasets while keeping the feature extraction network of the source model intact. As we can see, removing data classes from the source ImageNet dataset can lead to improved transfer learning accuracy compared to the conventional method of using the pre-trained model on the full ImageNet (*i.e.*, $n = 0$). Moreover, our proposed

$\ell_1$-sparse MU method achieves comparable or even slightly better transfer learning accuracy than the retraining-based approach [51]. Importantly, $\ell_1$-sparse MU offers the advantage of computational efficiency $2\times$ speed up over previous method [51] across all cases, making it an appealing choice for transfer learning using large-scale models. Here we remark that in order to align with previous method [51], we employed a fast-forward computer vision training pipeline (FFCV) [54] to accelerate our ImageNet training on GPUs.

**Additional results.** We found that model sparsity also enhances the privacy of the unlearned model, as evidenced by a lower MIA-Privacy. Refer to Appendix C.4 and Fig. A1 for more results. In addition,

Table 4: Transfer learning accuracy (Acc) and computation time (mins) of the unlearned ImageNet model with $n \in \{100, 200, 300\}$ classes removed, where SUN397 and OxfordPets are downstream target datasets on linear probing transfer learning setting. When $n = 0$, transfer learning is performed using the pretrained model on the full ImageNet, serving as a baseline, together with the method in [51] for comparison.

| Forgetting class # | 0 | 100 | | 200 | | 300 | |
|---|---|---|---|---|---|---|---|
| | Acc | Acc | Time | Acc | Time | Acc | Time |
| OxfordPets | | | | | | | |
| Method [51] | 85.70 | 85.79 | 71.84 | 86.10 | 61.53 | 86.32 | 54.53 |
| $\ell_1$-sparse MU | | 85.83 | 35.47 | 86.12 | 30.19 | 86.26 | 26.49 |
| SUN397 | | | | | | | |
| Method [51] | 46.55 | 46.97 | 73.26 | 47.14 | 61.43 | 47.31 | 55.24 |
| $\ell_1$-sparse MU | | 47.20 | 36.69 | 47.25 | 30.96 | 47.37 | 27.12 |

we have expanded our experimental scope to encompass the 'prune first, then unlearn' approach across various datasets and architectures. The results can be found in Tab. A3, Tab. A4, and Tab. A5. Furthermore, we conducted experiments on the $\ell_1$-sparse MU across different datasets, the Swin-Transformer architecture, and varying model sizes within the ResNet family. The corresponding findings are presented in Fig. A2 and Tab. A6, A7, A8 and A9.

# 6  Related Work

While Sec. 2 provides a summary of related works concerning exact and approximate unlearning methods and metrics, a more comprehensive review is provided below.

**Machine unlearning.** In addition to exact and approximate unlearning methods as we have reviewed in Sec. 2, there exists other literature aiming to develop the probabilistic notion of unlearning [35, 55–58], in particular through the lens of differential privacy (DP) [59]. Although DP enables unlearning with provable error guarantees, they typically require strong model and algorithmic assumptions and could lack effectiveness when facing practical adversaries, *e.g.*, membership inference attacks. Indeed, evaluating MU is far from trivial [8, 9, 11]. Furthermore, the attention on MU has also been raised

in different learning paradigms, *e.g.*, federated learning [29, 60], graph neural networks [61–63], and adversarial ML [64, 65]. In addition to preventing the leakage of data privacy from the trained models, the concept of MU has also inspired other emergent applications such as adversarial defense against backdoor attacks [6, 50] that we have studied and erasing image concepts of conditional generative models [66, 67].

**Understanding data influence.** The majority of MU studies are motivated by data privacy. Yet, they also closely relate to another line of research on understanding data influence in ML. For example, the influence function approach [32] has been used as an algorithmic backbone of many unlearning methods [6, 10]. From the viewpoint of data influence, MU has been used in the use case of adversarial defense against data poisoning backdoor attacks [50]. Beyond unlearning, evaluation of data influence has also been studied in fair learning [68, 69], transfer learning [51], and dataset pruning [70, 71].

**Model pruning.** The deployment constraints on *e.g.*, computation, energy, and memory necessitate the pruning of today's ML models, *i.e.*, promoting their weight sparsity. The vast majority of existing works [13–19] focus on developing model pruning methods that can strike a graceful balance between model's generalization and sparsity. In particular, the existence of LTH (lottery ticket hypothesis) [15] demonstrates the feasibility of co-improving the model's generalization and efficiency (in terms of sparsity) [40, 72–75]. In addition to generalization, model sparsity achieved by pruning can also be leveraged to improve other performance metrics, such as robustness [20–22], model explanation [25, 26], and privacy [27, 28, 76, 77].

# 7 Conclusion

In this work, we advance the method of machine unlearning through a novel viewpoint: model sparsification, achieved by weight pruning. We show in both theory and practice that model sparsity plays a foundational and crucial role in closing the gap between exact unlearning and existing approximate unlearning methods. Inspired by that, we propose two new unlearning paradigms, 'prune first, then unlearn' and 'sparsity-aware unlearn', which can significantly improve the efficacy of approximate unlearning. We demonstrate the effectiveness of our findings and proposals in extensive experiments across different unlearning setups. Our study also indicates the presence of *model modularity* traits, such as weight sparsity, that could simplify the process of machine unlearning. This may open up exciting prospects for future research to investigate unlearning patterns within weight or architecture space.

# 8 Acknowledgement

The work of J. Jia, J. Liu, Y. Yao, and S. Liu were supported by the Cisco Research Award and partially supported by the NSF Grant IIS-2207052, and the ARO Award W911NF2310343. Y. Liu was partially supported by NSF Grant IIS-2143895 and IIS-2040800.

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

# Appendix

## A   Proof of Proposition 1

Recap the definition of model update $\Delta(\mathbf{w})$ in (1) and $\boldsymbol{\theta}_\mathrm{o} = \boldsymbol{\theta}(\mathbf{1}/N)$, we approximate $\Delta(\mathbf{w})$ by the first-order Taylor expansion of $\boldsymbol{\theta}(\mathbf{w})$ at $\mathbf{w} = \mathbf{1}/N$. This leads to

$$\Delta(\mathbf{w}) = \boldsymbol{\theta}(\mathbf{w}) - \boldsymbol{\theta}(\mathbf{1}/N) \approx \left. \frac{d\boldsymbol{\theta}(\mathbf{w})}{d\mathbf{w}} \right|_{\mathbf{w}=\mathbf{1}/N} (\mathbf{w} - \mathbf{1}/N), \tag{A1}$$

where $\frac{d\boldsymbol{\theta}(\mathbf{w})}{d\mathbf{w}} \in \mathbb{R}^{M \times N}$, and recall that $M = |\boldsymbol{\theta}_\mathrm{o}|$ is the number of model parameters. The gradient $\frac{d\boldsymbol{\theta}(\mathbf{w})}{d\mathbf{w}}$ is known as implicit gradient [78] since it is defined through the solution of the optimization problem $\boldsymbol{\theta}(\mathbf{w}) = \arg\min_{\boldsymbol{\theta}} L(\mathbf{w}, \boldsymbol{\theta})$, where recall that $L(\mathbf{w}, \boldsymbol{\theta}) = \sum_{i=1}^{N} [w_i \ell_i(\boldsymbol{\theta}, \mathbf{z}_i)]$. By the stationary condition of $\boldsymbol{\theta}(\mathbf{w})$, we obtain

$$\nabla_{\boldsymbol{\theta}} L(\mathbf{w}, \boldsymbol{\theta}(\mathbf{w})) = \mathbf{0}. \tag{A2}$$

Next, we take the derivative of (A2) w.r.t. $\mathbf{w}$ based on the implicit function theorem [78] assuming that $\boldsymbol{\theta}(\mathbf{w})$ is the unique solution to minimizing $L$. This leads to

$$\left[ \frac{d\boldsymbol{\theta}(\mathbf{w})}{d\mathbf{w}} \right]^T \left[ \nabla_{\boldsymbol{\theta},\boldsymbol{\theta}} L(\mathbf{w}, \boldsymbol{\theta})|_{\boldsymbol{\theta}=\boldsymbol{\theta}(\mathbf{w})} \right] + \nabla_{\mathbf{w},\boldsymbol{\theta}} L(\mathbf{w}, \boldsymbol{\theta}(\mathbf{w})) = \mathbf{0}, \tag{A3}$$

where $\nabla_{\mathbf{a},\mathbf{b}} = \nabla_{\mathbf{a}} \nabla_{\mathbf{b}} \in \mathbb{R}^{|\mathbf{a}| \times |\mathbf{b}|}$ is the second-order partial derivative. Therefore,

$$\frac{d\boldsymbol{\theta}(\mathbf{w})}{d\mathbf{w}} = - \left[ \nabla_{\boldsymbol{\theta},\boldsymbol{\theta}} L(\mathbf{w}, \boldsymbol{\theta}(\mathbf{w})) \right]^{-1} \nabla_{\mathbf{w},\boldsymbol{\theta}} L(\mathbf{w}, \boldsymbol{\theta}(\mathbf{w}))^T, \tag{A4}$$

where $\nabla_{\mathbf{w},\boldsymbol{\theta}} L(\mathbf{w}, \boldsymbol{\theta}(\mathbf{w}))$ can be expanded as

$$\nabla_{\mathbf{w},\boldsymbol{\theta}} L(\mathbf{w}, \boldsymbol{\theta}(\mathbf{w})) = \nabla_{\mathbf{w}} \nabla_{\boldsymbol{\theta}} \sum_{i=1}^{N} [w_i \ell_i(\boldsymbol{\theta}(\mathbf{w}), \mathbf{z}_i)] \tag{A5}$$

$$= \nabla_{\mathbf{w}} \sum_{i=1}^{N} [w_i \nabla_{\boldsymbol{\theta}} \ell_i(\boldsymbol{\theta}(\mathbf{w}), \mathbf{z}_i)] \tag{A6}$$

$$= \begin{bmatrix} \nabla_{\boldsymbol{\theta}} \ell_1(\boldsymbol{\theta}(\mathbf{w}), \mathbf{z}_1)^T \\ \nabla_{\boldsymbol{\theta}} \ell_2(\boldsymbol{\theta}(\mathbf{w}), \mathbf{z}_2)^T \\ \vdots \\ \nabla_{\boldsymbol{\theta}} \ell_N(\boldsymbol{\theta}(\mathbf{w}), \mathbf{z}_N)^T \end{bmatrix}. \tag{A7}$$

Based on (A4) and (A7), we obtain the closed-form of implicit gradient at $\mathbf{w} = \mathbf{1}/N$:

$$\frac{d\boldsymbol{\theta}(\mathbf{w})}{d\mathbf{w}} |_{\mathbf{w}=\mathbf{1}/N} = - \left[ \nabla_{\boldsymbol{\theta},\boldsymbol{\theta}} L(\mathbf{1}/N, \boldsymbol{\theta}(\mathbf{1}/N)) \right]^{-1} \left[ \nabla_{\boldsymbol{\theta}} \ell_1(\boldsymbol{\theta}(\mathbf{1}/N), \mathbf{z}_1) \quad \dots \quad \nabla_{\boldsymbol{\theta}} \ell_N(\boldsymbol{\theta}(\mathbf{1}/N), \mathbf{z}_N) \right]$$

$$= - \mathbf{H}^{-1} \left[ \nabla_{\boldsymbol{\theta}} \ell_1(\boldsymbol{\theta}(\mathbf{1}/N), \mathbf{z}_1) \quad \dots \quad \nabla_{\boldsymbol{\theta}} \ell_N(\boldsymbol{\theta}(\mathbf{1}/N), \mathbf{z}_N) \right], \tag{A8}$$

where $\mathbf{H} = \nabla_{\boldsymbol{\theta},\boldsymbol{\theta}} L(\mathbf{1}/N, \boldsymbol{\theta}(\mathbf{1}/N))$.

Substituting (A8) into (A1), we obtain

$$\Delta(\mathbf{w}) \approx - \mathbf{H}^{-1} \left[ \nabla_{\boldsymbol{\theta}} \ell_1(\boldsymbol{\theta}(\mathbf{1}/N), \mathbf{z}_1) \quad \dots \quad \nabla_{\boldsymbol{\theta}} \ell_N(\boldsymbol{\theta}(\mathbf{1}/N), \mathbf{z}_N) \right] (\mathbf{w} - \mathbf{1}/N)$$

$$= - \mathbf{H}^{-1} \sum_{i=1}^{N} [(w_i - 1/N) \nabla_{\boldsymbol{\theta}} \ell_i(\boldsymbol{\theta}(\mathbf{1}/N), \mathbf{z}_i)]$$

$$= \mathbf{H}^{-1} \nabla_{\boldsymbol{\theta}} L(\mathbf{1}/N - \mathbf{w}, \boldsymbol{\theta}_\mathrm{o}), \tag{A9}$$

where the last equality holds by the definition of $L(\mathbf{w}, \boldsymbol{\theta}) = \sum_{i=1}^{N} [w_i \ell_i(\boldsymbol{\theta}, \mathbf{z}_i)]$.

The proof is now complete. $\qquad\square$

**Remark on IU using ave-ERM vs. sum-ERM.** Recall that the weighted empirical risk minimization (ERM) loss used in proposition (1), $L(\mathbf{w}, \boldsymbol{\theta}) = \sum_{i=1}^{N}[w_i \ell_i(\boldsymbol{\theta}, \mathbf{z}_i)]$ corresponds to the ave-ERM as $\mathbf{w}$ is subject to the simplex constraint ($\mathbf{1}^T \mathbf{w} = 1$ and $\mathbf{w} \geq \mathbf{0}$). This is different from the conventional derivation of IU using the sum-ERM [32, 35] in the absence of simplex constraint. In what follows, we discuss the impact of ave-ERM on IU vs. sum-ERM.

Noting that $\boldsymbol{\theta}_o$ represents the original model trained through conventional ERM, namely, the weighted ERM loss by setting $w_i = c$ ($\forall i$) for a positive constant $c$. Given the unlearning scheme (encoded in $\mathbf{w}$), the IU approach aims to delineate the model parameter adjustments required by MU from the initial model $\boldsymbol{\theta}_o$. Such a model weight modification is represented as $\Delta(\mathbf{w}) = \boldsymbol{\theta}(\mathbf{w}) - \boldsymbol{\theta}_o$ mentioned in proposition (1). The difference between ave-ERM and sum-ERM would play a role in deriving $\Delta(\mathbf{w})$, which relies on the Taylor expansion of $\boldsymbol{\theta}(\mathbf{w})$ (viewed as a function of $\mathbf{w}$). When the sum-ERM is considered, then the linearization point is typically set by $\mathbf{w} = \mathbf{1}$. This leads to

$$\Delta^{(\text{sum})}(\mathbf{w}) = \boldsymbol{\theta}(\mathbf{w}) - \boldsymbol{\theta}(\mathbf{1}) \approx \boldsymbol{\theta}(\mathbf{1}) + \frac{d\boldsymbol{\theta}(\mathbf{w})}{d\mathbf{w}}|_{\mathbf{w}=\mathbf{1}}(\mathbf{w} - \mathbf{1}) - \boldsymbol{\theta}(\mathbf{1})$$
$$= \frac{d\boldsymbol{\theta}(\mathbf{w})}{d\mathbf{w}}|_{\mathbf{w}=\mathbf{1}}(\mathbf{w} - \mathbf{1}), \tag{A10}$$

where $\boldsymbol{\theta}(\mathbf{1}) = \boldsymbol{\theta}_o$ for sum-ERM, and $\frac{d\boldsymbol{\theta}(\mathbf{w})}{d\mathbf{w}}$ is implicit gradient [78] since it is defined upon an implicit optimization problem $\boldsymbol{\theta}(\mathbf{w}) = \arg\min_{\boldsymbol{\theta}} L(\mathbf{w}, \boldsymbol{\theta})$.

When the ave-ERM is considered, the linearization point is set by $\mathbf{w} = \mathbf{1}/N$. This leads to

$$\Delta^{(\text{ave})}(\mathbf{w}) = \boldsymbol{\theta}(\mathbf{w}) - \boldsymbol{\theta}(\mathbf{1}/N) \approx \boldsymbol{\theta}(\mathbf{1}/N) + \frac{d\boldsymbol{\theta}(\mathbf{w})}{d\mathbf{w}}|_{\mathbf{w}=\mathbf{1}/N}(\mathbf{w} - \mathbf{1}/N) - \boldsymbol{\theta}(\mathbf{1}/N)$$
$$= \frac{d\boldsymbol{\theta}(\mathbf{w})}{d\mathbf{w}}|_{\mathbf{w}=\mathbf{1}/N}(\mathbf{w} - \mathbf{1}/N), \tag{A11}$$

where $\boldsymbol{\theta}(\mathbf{1}/N) = \boldsymbol{\theta}_o$ for ave-ERM. The derivation of the implicit gradient $\frac{d\boldsymbol{\theta}(\mathbf{w})}{d\mathbf{w}}$ is shown in (A8).

Next, let us draw a comparison between $\Delta^{(\text{sum})}(\mathbf{w})$ and $\Delta^{(\text{ave})}(\mathbf{w})$ using a specific example below. If we aim to unlearn the first $k$ training data points, the unlearning weights $\mathbf{w}_{\text{MU}}$ under sum-ERM is then given by $\mathbf{w}_{\text{MU}}^{(\text{sum})} = [\underbrace{0, 0, \ldots, 0}_{k \ 0s}, 1, 1, \ldots, 1]$, where 0 encodes the data sample to be unlearned or removed. This yields $(\mathbf{w}_{\text{MU}}^{(\text{sum})} - \mathbf{1}) = [\underbrace{1, 1, \ldots, 1}_{k \ 1s}, 0, 0, \ldots, 0]$. By contrast, the unlearning weights $\mathbf{w}_{\text{MU}}$ under ave-ERM is given by $\mathbf{w}_{\text{MU}}^{(\text{ave})} = [\underbrace{0, 0, \ldots, 0}_{k \ 0s}, \frac{1}{N-k}, \frac{1}{N-k}, \ldots, \frac{1}{N-k}]$. As a result,

$(\mathbf{w}_{\text{MU}}^{(\text{ave})} - \mathbf{1}/N) = [\underbrace{-\frac{1}{N}, -\frac{1}{N}, \ldots, -\frac{1}{N}}_{k \ \frac{1}{N}s}, \frac{1}{N-k} - \frac{1}{N}, \frac{1}{N-k} - \frac{1}{N}, \ldots, \frac{1}{N-k} - \frac{1}{N}]$. The above difference is caused by the presence of simplex constraint of $\mathbf{w}$ in ave-ERM. Thus, the MU's weight configuration $(\mathbf{w}_{\text{MU}}^{(\text{ave})} - \mathbf{1}/N)$ obtained from ave-ERM is different from $(\mathbf{w}_{\text{MU}}^{(\text{sum})} - \mathbf{1})$ in the sum-ERM setting.

Given the above example, the error term of the Taylor expansion using sum-ERM for $\mathbf{w} = \mathbf{w}_{\text{MU}}^{\text{sum}}$ is in the order of $\|\mathbf{w}_{\text{MU}}^{\text{sum}} - \mathbf{1}\|_2^2 = k$, while the error term using ave-ERM for $\mathbf{w} = \mathbf{w}_{\text{MU}}^{\text{ave}}$ is in the order of $\|\mathbf{w}_{\text{MU}}^{\text{ave}} - \mathbf{1}/N\|_2^2 = \frac{k}{N^2} + \frac{k^2}{N^2(N-k)} = \frac{k}{N(N-k)}$. Thus compared to ave-ERM, the use of sum-ERM could cause the first-order Taylor expansion in IU less accurate as the number of unlearning datapoints ($k$) increases. Furthermore, the IG $\frac{d\boldsymbol{\theta}(\mathbf{w})}{d\mathbf{w}}|_{\mathbf{w}=\mathbf{1}}$ in sum-ERM is also different from $\frac{d\boldsymbol{\theta}(\mathbf{w})}{d\mathbf{w}}|_{\mathbf{w}=\mathbf{1}/N}$ in ave-ERM as they are evaluated at two different linearization points.

## B   Proof of Proposition 2

The proof follows [8, Sec. 5], with the additional condition that the model is **sparse** encoded by a pre-fixed (binary) pruning mask $\mathbf{m}$, namely, $\boldsymbol{\theta}' := \mathbf{m} \odot \boldsymbol{\theta}$. Then, based on [8, Eq. 5], the model

updated by SGD yields

$$\boldsymbol{\theta}'_t \approx \boldsymbol{\theta}'_0 - \eta \mathbf{m} \odot \sum_{i=1}^{t-1} \nabla_{\boldsymbol{\theta}} \ell(\boldsymbol{\theta}'_0, \hat{\mathbf{z}}_i) + \mathbf{m} \odot \left( \sum_{i=1}^{t-1} f(i) \right), \tag{A12}$$

where $\boldsymbol{\theta}'_0 = \mathbf{m} \odot \boldsymbol{\theta}_0$ is the model initialization when using SGD-based sparse training, $\{\hat{\mathbf{z}}_i\}$ is the sequence of stochastic data samples, $t$ the number of training iterations, $\eta$ is the learning rate, and $f(i)$ is defined recursively as

$$f(i) = -\eta \nabla^2_{\boldsymbol{\theta}, \boldsymbol{\theta}} \ell(\boldsymbol{\theta}'_0, \hat{\mathbf{z}}_i) \left( -\eta \sum_{j=0}^{i-1} \mathbf{m} \odot \nabla_{\boldsymbol{\theta}} \ell(\boldsymbol{\theta}'_0, \hat{\mathbf{z}}_j) + \sum_{j=0}^{i-1} (\mathbf{m} \odot f(j)) \right), \tag{A13}$$

with $f(0) = 0$. Inspired by the second term of (A12), to unlearn the data sample $\hat{\mathbf{z}}_i$, we will have to add back the first-order gradients under $\hat{\mathbf{z}}_i$. This corresponds to the GA-based approximate unlearning method. Yet, this approximate unlearning introduces an unlearning error, given by the last term of (A12)

$$\mathbf{e}_{\mathbf{m}}(\boldsymbol{\theta}_0, \{\hat{\mathbf{z}}_i\}, t, \eta) := \mathbf{m} \odot \left( \sum_{i=1}^{t-1} f(i) \right). \tag{A14}$$

Next, if we interpret the mask $\mathbf{m}$ as a diagonal matrix $\mathrm{diag}(\mathbf{m})$ with 0's and 1's along its diagonal based on $\mathbf{m}$, we can then express the sparse model $\mathbf{m} \odot \boldsymbol{\theta}$ as $\mathrm{diag}(\mathbf{m})\boldsymbol{\theta}$. Similar to [8, Eq. 9], we can derive a bound on the unlearning error (A14) by ignoring the terms other than those with $\eta^2$ in $f(i)$, *i.e.*, (A13). This is because, in the recursive form of $f(i)$, all other terms exhibit a higher degree of the learning rate $\eta$ compared to $\eta^2$. As a result, we obtain

$$e(\mathbf{m}) = \|\mathbf{e}_{\mathbf{m}}(\boldsymbol{\theta}_0, \{\hat{\mathbf{z}}_i\}, t, \eta)\|_2 = \left\| \mathbf{m} \odot \left( \sum_{i=1}^{t-1} f(i) \right) \right\|_2$$

$$\approx \eta^2 \left\| \mathrm{diag}(\mathbf{m}) \sum_{i=1}^{t-1} \nabla^2_{\boldsymbol{\theta}, \boldsymbol{\theta}} \ell(\boldsymbol{\theta}'_0, \hat{\mathbf{z}}_i) \sum_{j=0}^{i-1} \mathbf{m} \odot \nabla_{\boldsymbol{\theta}} \ell(\boldsymbol{\theta}'_0, \hat{\mathbf{z}}_j) \right\|_2$$

$$\leq \eta^2 \sum_{i=1}^{t-1} \left\| \mathrm{diag}(\mathbf{m}) \nabla^2_{\boldsymbol{\theta}, \boldsymbol{\theta}} \ell(\boldsymbol{\theta}'_0, \hat{\mathbf{z}}_i) \sum_{j=0}^{i-1} \mathbf{m} \odot \nabla_{\boldsymbol{\theta}} \ell(\boldsymbol{\theta}'_0, \hat{\mathbf{z}}_j) \right\|_2 \quad \text{(Triangle inequality)}$$

$$\leq \eta^2 \sum_{i=1}^{t-1} \left\| \mathrm{diag}(\mathbf{m}) \nabla^2_{\boldsymbol{\theta}, \boldsymbol{\theta}} \ell(\boldsymbol{\theta}'_0, \hat{\mathbf{z}}_i) \right\| \left\| \sum_{j=0}^{i-1} \mathbf{m} \odot \nabla_{\boldsymbol{\theta}} \ell(\boldsymbol{\theta}'_0, \hat{\mathbf{z}}_j) \right\|_2 \tag{A15}$$

$$\lesssim \eta^2 \sum_{i=1}^{t-1} \left\| \mathrm{diag}(\mathbf{m}) \nabla^2_{\boldsymbol{\theta}, \boldsymbol{\theta}} \ell(\boldsymbol{\theta}'_0, \hat{\mathbf{z}}_i) \right\| \frac{i}{t} \left\| \boldsymbol{\theta}'_t - \boldsymbol{\theta}'_0 \right\|_2 \tag{A16}$$

$$\leq \eta^2 \sigma(\mathbf{m}) \|\mathbf{m} \odot (\boldsymbol{\theta}_t - \boldsymbol{\theta}_0)\|_2 \frac{1}{t} \frac{t-1}{2} t = \frac{\eta^2}{2}(t-1)\|\mathbf{m} \odot (\boldsymbol{\theta}_t - \boldsymbol{\theta}_0)\|_2 \sigma(\mathbf{m}), \tag{A17}$$

where the inequality (A16) holds given the fact that $\sum_{j=0}^{i-1} \mathbf{m} \odot \nabla_{\boldsymbol{\theta}} \ell(\boldsymbol{\theta}'_0, \hat{\mathbf{z}}_j)$ in (A15) can be approximated by its expectation $\frac{i(\boldsymbol{\theta}'_t - \boldsymbol{\theta}'_0)}{t}$ [8, Eq. 7], and $\sigma(\mathbf{m}) := \max_j \{\sigma_j(\nabla^2_{\boldsymbol{\theta}, \boldsymbol{\theta}} \ell), \text{if } m_j \neq 0\}$, *i.e.*, the largest eigenvalue among the dimensions left intact by the binary mask $\mathbf{m}$. The above suggests that the unlearning error might be large if $\mathbf{m} = \mathbf{1}$ (no pruning). Based on (A17), we can then readily obtain the big $O$ notation in (2). This completes the proof.

## C  Additional Experimental Details and Results

### C.1  Datasets and models

We summarize the datasets and model configurations in Tab. A1.

Table A1: Dataset and model setups.

| Settings | CIFAR-10 | | SVHN | CIFAR-100 | ImageNet |
| | ResNet-18 | VGG-16 | ResNet-18 | ResNet-18 | ResNet-18 |
| --- | --- | --- | --- | --- | --- |
| Batch Size | 256 | 256 | 256 | 256 | 1024 |

Table A2: Detailed training details for model pruning.

| Experiments | CIFAR-10/CIFAR-100 | SVHN | ImageNet |
| --- | --- | --- | --- |
| Training epochs | 182 | 160 | 90 |
| Rewinding epochs | 8 | 8 | 5 |
| Momentum | 0.9 | 0.9 | 0.875 |
| $\ell_2$ regularization | $5e^{-4}$ | $5e^{-4}$ | $3.05e^{-5}$ |
| Warm-up epochs | 1(75 for VGG-16) | 0 | 8 |

## C.2 Additional training and unlearning settings

**Training configuration of pruning.** For all pruning methods, including IMP [15], SynFlow [40], and OMP [17], we adopt the settings from the current SOTA implementations [17]; see a summary in Tab. A2. For IMP, OMP, and SynFlow, we adopt the step learning rate scheduler with a decay rate of 0.1 at 50% and 75% epochs. We adopt 0.1 as the initial learning rate for all pruning methods.

**Additional training details of MU.** For all datasets and model architectures, we adopt 10 epochs for FT, and 5 epochs for GA method. The learning rate for FT and GA are carefully tuned between $[10^{-5}, 0.1]$ for each dataset and model architecture. In particular, we adopt 0.01 as the learning rate for FT method and $10^{-4}$ for GA on the CIFAR-10 dataset (ResNet-18, class-wise forgetting) at different sparsity levels. By default, we choose SGD as the optimizer for the FT and GA methods. As for FF method, we perform a greedy search for hyperparameter tuning [12] between $10^{-9}$ and $10^{-6}$.

## C.3 Detailed metric settings

**Details of MIA implementation.** MIA is implemented using the prediction confidence-based attack method [48]. There are mainly two phases during its computation: **(1) training phase**, and **(2) testing phase**. To train an MIA model, we first sample a balanced dataset from the remaining dataset ($\mathcal{D}_r$) and the test dataset (different from the forgetting dataset $\mathcal{D}_f$) to train the MIA predictor. The learned MIA is then used for MU evaluation in its testing phase. To evaluate the performance of MU, MIA-Efficacy is obtained by applying the learned MIA predictor to the unlearned model ($\boldsymbol{\theta}_u$) on the forgetting dataset ($\mathcal{D}_f$). Our objective is to find out how many samples in $\mathcal{D}_f$ can be correctly predicted as non-training samples by the MIA model against $\boldsymbol{\theta}_u$. The formal definition of MIA-Efficacy is then given by:

$$\text{MIA-Efficacy} = \frac{TN}{|\mathcal{D}_f|}, \tag{A18}$$

where $TN$ refers to the true negatives predicted by our MIA predictor, *i.e.*, the number of the forgetting samples predicted as non-training examples, and $|\mathcal{D}_f|$ refers to the size of the forgetting dataset. As described above, MIA-Efficacy leverages the privacy attack to justify how good the unlearning performance could be.

## C.4 Additional experiment results

**Model sparsity benefits privacy of MU for 'free'.** It was recently shown in [27, 28] that model sparsification helps protect data privacy, in terms of defense against MIA used to infer training data information from a learned model. Inspired by the above, we ask if sparsity can also bring the privacy benefit to an unlearned model, evaluated by the MIA rate on the remaining dataset $\mathcal{D}_r$ (that we term **MIA-Privacy**). This is different from MIA-Efficacy, which reflects the efficacy of scrubbing $\mathcal{D}_f$, *i.e.*, correctly predicting that data sample in $\mathcal{D}_f$ is not in the training set of the unlearned model. In contrast, MIA-Privacy characterizes the *privacy* of the unlearned model about $\mathcal{D}_r$. A *lower* MIA-Privacy implies *less* information leakage.

**Fig. A1** shows MIA-Privacy of unlearned models versus the sparsity ratio applied to different unlearning methods in the 'prune first, then unlearn' paradigm. As we can see, MIA-Privacy decreases as the sparsity increases. This suggests the improved privacy of unlearning on sparse models. Moreover, we observe that approximate unlearning outperforms exact unlearning (Retrain) in privacy preservation of $\mathcal{D}_r$. This is because Retrain is conducted over $\mathcal{D}_r$ from scratch, leading to the strongest dependence on $\mathcal{D}_r$ than other unlearning methods. Another interesting observation is that IU and GA yield a much smaller MIA-Privacy than other approximate unlearning methods. The rationale behind that is IU and GA have a weaker correlation with $\mathcal{D}_r$ during unlearning. Specifically, the unlearning loss of IU only involves the forgetting data influence weights, *i.e.*, $(\mathbf{1}/N - \mathbf{w})$ in (1). Similarly, GA only performs gradient ascent over $\mathcal{D}_f$, with the least dependence on $\mathcal{D}_r$.

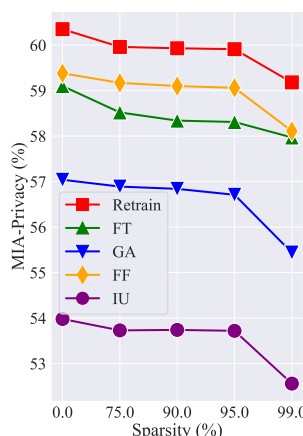

Figure A1: Privacy on $\mathcal{D}_r$ (MIA-Privacy) using different unlearning methods vs. model sparsity.

**Performance of 'prune first, then unlearn' on various datasets and architectures.** As demonstrated in Tab. A3 and Tab. A4, the introduction of model sparsity can effectively reduce the discrepancy between approximate and exact unlearning across a diverse range of datasets and architectures. This phenomenon is observable in various unlearning scenarios. Remarkably, model sparsity enhances both UA and MIA-Efficacy metrics without incurring substantial degradation on RA and TA in different unlearning scenarios. These observations corroborate the findings reported in Tab. 3.

Table A3: MU performance vs. sparsity on additional datasets (CIFAR-100 [43] and SVHN [45]) for both class-wise forgetting and random data forgetting. The content format follows Tab. 3.

| MU | UA DENSE | UA 95% Sparsity | MIA-Efficacy DENSE | MIA-Efficacy 95% Sparsity | RA DENSE | RA 95% Sparsity | TA DENSE | TA 95% Sparsity | RTE (min) |
|---|---|---|---|---|---|---|---|---|---|
| *Class-wise forgetting, CIFAR-100* | | | | | | | | | |
| Retrain | $100.00_{\pm0.00}$ | $100.00_{\pm0.00}$ | $100.00_{\pm0.00}$ | $100.00_{\pm0.00}$ | $99.97_{\pm0.01}$ | $96.68_{\pm0.15}$ | $73.74_{\pm0.19}$ | $69.49_{\pm0.41}$ | 48.45 |
| FT | $26.45_{\pm6.29}(73.55)$ | $73.63_{\pm5.06}(26.37)$ | $92.44_{\pm5.93}(7.56)$ | $98.88_{\pm4.32}(1.12)$ | $99.86_{\pm0.04}(0.11)$ | $97.72_{\pm0.47}(1.04)$ | $74.08_{\pm0.23}(0.74)$ | $71.37_{\pm0.18}(3.00)$ | 3.76 |
| GA | $81.47_{\pm0.32}(18.53)$ | $99.01_{\pm0.01}(0.99)$ | $93.47_{\pm4.56}(6.53)$ | $100.00_{\pm0.00}(0.00)$ | $90.33_{\pm1.71}(9.64)$ | $80.45_{\pm0.78}(16.23)$ | $64.94_{\pm0.74}(8.80)$ | $60.99_{\pm0.14}(8.50)$ | 0.21 |
| IU | $84.12_{\pm0.34}(15.88)$ | $99.78_{\pm0.01}(0.22)$ | $98.44_{\pm0.45}(1.56)$ | $99.33_{\pm0.00}(0.67)$ | $96.23_{\pm0.02}(3.74)$ | $95.45_{\pm0.17}(1.23)$ | $71.24_{\pm0.22}(2.50)$ | $70.79_{\pm0.11}(0.95)$ | 4.30 |
| *Random data forgetting, CIFAR-100* | | | | | | | | | |
| Retrain | $24.76_{\pm0.12}$ | $27.64_{\pm1.03}$ | $49.80_{\pm0.26}$ | $44.87_{\pm0.81}$ | $99.98_{\pm0.02}$ | $99.24_{\pm0.02}$ | $74.46_{\pm0.08}$ | $69.78_{\pm0.15}$ | 48.70 |
| FT | $0.78_{\pm0.34}(23.98)$ | $8.37_{\pm1.63}(19.27)$ | $1.13_{\pm0.40}(48.67)$ | $18.57_{\pm1.57}(26.30)$ | $99.93_{\pm0.02}(0.05)$ | $99.20_{\pm0.27}(0.04)$ | $75.14_{\pm0.09}(0.68)$ | $73.18_{\pm0.30}(1.60)$ | 3.74 |
| GA | $0.04_{\pm0.02}(24.75)$ | $3.92_{\pm0.28}(23.72)$ | $3.80_{\pm0.87}(46.00)$ | $7.51_{\pm1.37}(37.36)$ | $99.97_{\pm0.01}(0.01)$ | $98.40_{\pm1.22}(0.84)$ | $74.07_{\pm0.11}(0.39)$ | $72.19_{\pm0.15}(2.41)$ | 0.24 |
| IU | $1.53_{\pm0.36}(23.23)$ | $6.01_{\pm0.17}(21.63)$ | $6.58_{\pm0.42}(43.22)$ | $11.47_{\pm0.54}(33.40)$ | $99.01_{\pm0.28}(0.97)$ | $96.53_{\pm0.24}(2.71)$ | $71.76_{\pm0.31}(2.70)$ | $69.40_{\pm0.19}(0.38)$ | 3.80 |
| *Class-wise forgetting, SVHN* | | | | | | | | | |
| Retrain | $100.00_{\pm0.00}$ | $100.00_{\pm0.00}$ | $100.00_{\pm0.00}$ | $100.00_{\pm0.00}$ | $100.00_{\pm0.00}$ | $100.00_{\pm0.00}$ | $95.71_{\pm0.12}$ | $94.95_{\pm0.05}$ | 42.84 |
| FT | $11.48_{\pm8.12}(88.52)$ | $51.93_{\pm19.62}(48.07)$ | $86.12_{\pm9.62}(13.88)$ | $99.42_{\pm0.51}(0.58)$ | $100.00_{\pm0.00}(0.00)$ | $99.00_{\pm0.00}(1.00)$ | $95.99_{\pm0.07}(0.28)$ | $95.89_{\pm0.02}(0.94)$ | 2.86 |
| GA | $83.87_{\pm0.19}(16.13)$ | $86.52_{\pm0.11}(13.48)$ | $99.97_{\pm0.02}(0.03)$ | $100.00_{\pm0.00}(0.00)$ | $99.60_{\pm0.15}(0.40)$ | $98.37_{\pm0.11}(1.63)$ | $95.27_{\pm0.02}(0.44)$ | $93.42_{\pm0.07}(1.53)$ | 0.28 |
| IU | $95.11_{\pm0.02}(4.89)$ | $100.00_{\pm0.00}(0.00)$ | $99.89_{\pm0.04}(0.11)$ | $100.00_{\pm0.00}(0.00)$ | $100.00_{\pm0.00}(0.00)$ | $99.85_{\pm0.02}(0.15)$ | $95.70_{\pm0.09}(0.01)$ | $94.90_{\pm0.04}(0.05)$ | 3.19 |
| *Random data forgetting, SVHN* | | | | | | | | | |
| Retrain | $4.89_{\pm0.11}$ | $4.78_{\pm0.23}$ | $15.38_{\pm0.14}$ | $15.25_{\pm0.18}$ | $100.00_{\pm0.00}$ | $100.00_{\pm0.00}$ | $95.54_{\pm0.09}$ | $95.44_{\pm0.12}$ | 42.71 |
| FT | $3.56_{\pm0.27}(1.33)$ | $3.97_{\pm0.20}(0.81)$ | $10.05_{\pm0.24}(5.33)$ | $10.87_{\pm0.13}(4.38)$ | $99.89_{\pm0.04}(0.11)$ | $98.57_{\pm0.09}(1.43)$ | $93.55_{\pm0.12}(1.99)$ | $93.54_{\pm0.17}(1.90)$ | 2.73 |
| GA | $0.99_{\pm0.42}(3.90)$ | $2.68_{\pm0.53}(2.10)$ | $3.07_{\pm0.53}(12.31)$ | $9.31_{\pm0.48}(5.94)$ | $99.43_{\pm0.22}(0.57)$ | $97.83_{\pm0.43}(2.17)$ | $94.03_{\pm0.21}(1.51)$ | $93.33_{\pm0.27}(2.11)$ | 0.26 |
| IU | $3.48_{\pm0.13}(1.41)$ | $5.62_{\pm0.48}(0.84)$ | $9.44_{\pm0.27}(5.94)$ | $12.28_{\pm0.41}(2.97)$ | $96.30_{\pm0.08}(3.70)$ | $95.67_{\pm0.15}(4.33)$ | $91.59_{\pm0.11}(3.95)$ | $90.91_{\pm0.26}(4.53)$ | 3.21 |

Table A4: MU performance vs. sparsity on the additional architecture (VGG-16 [47]) for both class-wise forgetting and random data forgetting on CIFAR-10. The content format follows Tab. 3.

| MU | UA DENSE | UA 95% Sparsity | MIA-Efficacy DENSE | MIA-Efficacy 95% Sparsity | RA DENSE | RA 95% Sparsity | TA DENSE | TA 95% Sparsity | RTE (min) |
|---|---|---|---|---|---|---|---|---|---|
| *Class-wise forgetting, VGG-16* | | | | | | | | | |
| Retrain | $100.00_{\pm0.00}$ | $100.00_{\pm0.00}$ | $100.00_{\pm0.00}$ | $100.00_{\pm0.00}$ | $100.00_{\pm0.01}$ | $99.97_{\pm0.00}$ | $94.83_{\pm0.10}$ | $92.93_{\pm0.06}$ | 30.38 |
| FT | $28.00_{\pm8.16}(72.00)$ | $34.94_{\pm5.37}(65.06)$ | $63.23_{\pm17.68}(36.77)$ | $68.02_{\pm12.03}(31.98)$ | $99.87_{\pm0.05}(0.13)$ | $99.60_{\pm0.08}(0.37)$ | $92.80_{\pm1.28}(2.03)$ | $92.96_{\pm0.85}(0.03)$ | 1.81 |
| GA | $77.51_{\pm3.47}(22.49)$ | $83.93_{\pm2.14}(16.07)$ | $80.13_{\pm4.27}(19.87)$ | $88.04_{\pm3.18}(11.96)$ | $96.09_{\pm0.13}(3.91)$ | $97.33_{\pm0.08}(2.64)$ | $88.80_{\pm1.33}(6.03)$ | $89.95_{\pm0.78}(2.98)$ | 0.27 |
| IU | $88.58_{\pm0.86}(11.42)$ | $98.78_{\pm0.44}(1.22)$ | $92.27_{\pm1.14}(7.73)$ | $99.91_{\pm0.05}(0.09)$ | $96.89_{\pm0.27}(3.11)$ | $93.18_{\pm0.28}(6.79)$ | $89.81_{\pm1.01}(5.02)$ | $87.45_{\pm0.81}(5.48)$ | 2.51 |
| *Random data forgetting, VGG-16* | | | | | | | | | |
| Retrain | $7.13_{\pm0.60}$ | $7.47_{\pm0.30}$ | $13.02_{\pm0.77}$ | $13.51_{\pm0.50}$ | $100.00_{\pm0.01}$ | $99.93_{\pm0.01}$ | $92.80_{\pm0.17}$ | $91.98_{\pm0.22}$ | 30.29 |
| FT | $0.86_{\pm0.29}(6.27)$ | $1.46_{\pm0.22}(6.01)$ | $2.62_{\pm0.47}(10.40)$ | $3.82_{\pm0.41}(9.69)$ | $99.76_{\pm0.12}(0.24)$ | $99.47_{\pm0.11}(0.53)$ | $92.21_{\pm0.13}(0.59)$ | $92.03_{\pm0.37}(0.05)$ | 1.77 |
| GA | $9.11_{\pm0.83}(1.98)$ | $6.91_{\pm0.96}(0.56)$ | $7.77_{\pm1.01}(5.25)$ | $8.37_{\pm1.35}(5.14)$ | $93.08_{\pm0.93}(6.92)$ | $93.63_{\pm1.16}(6.30)$ | $86.44_{\pm1.32}(6.36)$ | $89.22_{\pm1.59}(4.53)$ | 0.31 |
| IU | $1.02_{\pm0.43}(6.11)$ | $3.07_{\pm0.50}(4.40)$ | $2.51_{\pm0.61}(9.51)$ | $6.86_{\pm0.67}(6.65)$ | $99.14_{\pm0.03}(0.86)$ | $97.35_{\pm0.31}(2.58)$ | $91.01_{\pm0.29}(1.79)$ | $89.49_{\pm0.37}(2.49)$ | 2.78 |

To demonstrate the effectiveness of our methods on a larger dataset, we conducted additional experiments on **ImageNet** [46] with settings consistent with the class-wise forgetting in Tab. 3. As we can see from Tab. A5, sparsity reduces the performance gap between exact unlearning (Retrain) and the approximate unlearning methods (FT and GA). The results are consistent with our observations in other datasets. Note that the 83% model sparsity (ImageNet, ResNet-18) is used to preserve the TA performance after one-shot magnitude (OMP) pruning.

**Performance of $\ell_1$ sparsity-aware MU on additional datasets.** As seen in Fig. A2, $\ell_1$-sparse MU significantly reduces the gap between approximate and exact unlearning methods across various

Table A5: Performance overview of MU vs. sparsity on ImageNet considering class-wise forgetting. The content format follows Tab. 3.

| MU | UA | | MIA-Efficacy | | RA | | TA | | RTE |
|---|---|---|---|---|---|---|---|---|---|
| | DENSE | 83% Sparsity | DENSE | 83% Sparsity | DENSE | 83% Sparsity | DENSE | 83% Sparsity | (hours) |
| Class-wise forgetting, ImageNet | | | | | | | | | |
| Retrain | 100.00 | 100.00 | 100.00 | 100.00 | 71.75 | 69.18 | 69.49 | 68.86 | 26.18 |
| FT | 63.60 (36.40) | 74.66 (25.34) | 68.61 (31.39) | 81.43 (18.57) | 72.45 (0.70) | 69.36 (0.18) | 69.80 (0.31) | 68.77 (0.09) | 2.87 |
| GA | 85.10 (14.90) | 90.21 (9.79) | 87.46 (12.54) | 94.25 (5.75) | 65.93 (5.82) | 62.94 (6.24) | 64.62 (4.87) | 64.65 (4.21) | 0.01 |

datasets (CIFAR-100 [43], SVHN [45], ImageNet [46]) in different unlearning scenarios. It notably outperforms other methods in UA and MIA-Efficacy metrics while preserving acceptable RA and TA performances, thus becoming a practical choice for unlearning scenarios. In class-wise and random data forgetting cases, $\ell_1$-sparse MU exhibits performance on par with Retrain in UA and MIA-Efficacy metrics. Importantly, the use of $\ell_1$-sparse MU consistently enhances forgetting metrics with an insignificant rise in computational cost compared with FT, underscoring its effectiveness and efficiency in diverse unlearning scenarios. For detailed numerical results, please refer to Tab. A6.

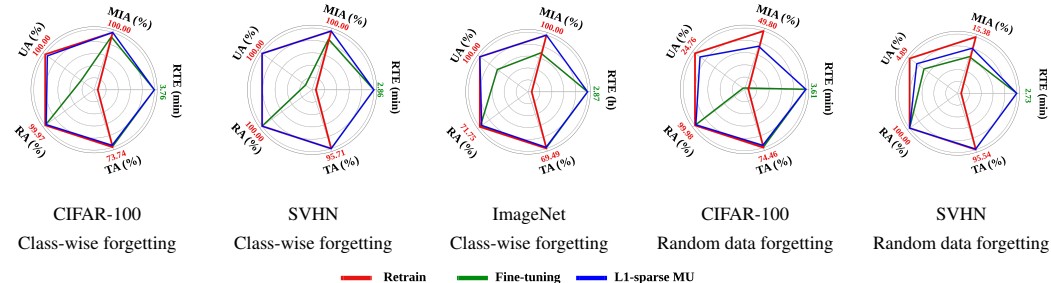

| CIFAR-100 | SVHN | ImageNet | CIFAR-100 | SVHN |
|---|---|---|---|---|
| Class-wise forgetting | Class-wise forgetting | Class-wise forgetting | Random data forgetting | Random data forgetting |

**— Retrain — Fine-tuning — L1-sparse MU**

Figure A2: Performance of sparsity-aware unlearning vs. FT and Retrain on class-wise forgetting and random data forgetting under ResNet-18 on different datasets. Each metric is normalized to $[0, 1]$ based on the best result across unlearning methods for ease of visualization, while the actual best value is provided. The figure format is consistent with Fig. 5.

Table A6: Performance of sparsity-aware MU vs. Retrain, FT, GA and IU considering class-wise forgetting and random data forgetting, where the table format and setup are consistent with Tab. 3. The unit of RTE is minutes for all datasets, except ImageNet. For ImageNet, indicated by an asterisk (∗), RTE is measured by hours.

| MU | UA | MIA-Efficacy | RA | TA | RTE (min) |
|---|---|---|---|---|---|
| Class-wise forgetting, CIFAR-10 | | | | | |
| Retrain | $100.00_{\pm0.00}$ | $100.00_{\pm0.00}$ | $100.00_{\pm0.00}$ | $94.83_{\pm0.11}$ | 43.23 |
| FT | $22.53_{\pm8.16}(77.47)$ | $75.00_{\pm14.68}(25.00)$ | $99.87_{\pm0.04}(0.13)$ | $94.31_{\pm0.19}(0.52)$ | 2.52 |
| GA | $93.08_{\pm2.29}(6.92)$ | $94.03_{\pm3.27}(5.97)$ | $92.60_{\pm0.25}(7.40)$ | $86.64_{\pm0.28}(8.19)$ | 0.33 |
| IU | $87.82_{\pm2.15}(12.18)$ | $95.96_{\pm0.21}(4.04)$ | $97.98_{\pm0.21}(2.02)$ | $91.42_{\pm0.21}(3.41)$ | 3.25 |
| $\ell_1$-sparse MU | $100.00_{\pm0.00}(0.00)$ | $100.00_{\pm0.00}(0.00)$ | $98.99_{\pm0.12}(1.01)$ | $93.40_{\pm0.43}(1.43)$ | 2.53 |
| Class-wise forgetting, CIFAR-100 | | | | | |
| Retrain | $100.00_{\pm0.00}$ | $100.00_{\pm0.00}$ | $99.97_{\pm0.01}$ | $73.74_{\pm0.19}$ | 48.45 |
| FT | $26.45_{\pm6.29}(73.55)$ | $92.44_{\pm5.93}(7.56)$ | $99.86_{\pm0.04}(0.11)$ | $74.08_{\pm0.23}(0.34)$ | 3.76 |
| GA | $81.47_{\pm0.32}(18.53)$ | $93.47_{\pm4.56}(6.53)$ | $90.33_{\pm1.71}(9.64)$ | $64.94_{\pm0.74}(8.80)$ | 0.21 |
| IU | $84.12_{\pm0.34}(15.88)$ | $98.44_{\pm0.45}(1.56)$ | $96.23_{\pm0.02}(3.74)$ | $71.24_{\pm0.22}(2.50)$ | 4.30 |
| $\ell_1$-sparse MU | $95.67_{\pm0.53}(4.33)$ | $100.00_{\pm0.00}(0.00)$ | $98.01_{\pm0.02}(1.96)$ | $71.35_{\pm0.22}(2.39)$ | 3.79 |
| Class-wise forgetting, SVHN | | | | | |
| Retrain | $100.00_{\pm0.00}$ | $100.00_{\pm0.00}$ | $100.00_{\pm0.00}$ | $95.71_{\pm0.12}$ | 42.84 |
| FT | $11.48_{\pm8.12}(88.52)$ | $86.12_{\pm9.62}(13.88)$ | $100.00_{\pm0.00}(0.00)$ | $95.99_{\pm0.07}(0.28)$ | 2.86 |
| GA | $83.87_{\pm0.19}(16.13)$ | $99.97_{\pm0.02}(0.03)$ | $99.60_{\pm0.15}(0.40)$ | $95.27_{\pm0.02}(0.44)$ | 0.28 |
| IU | $95.11_{\pm0.02}(4.89)$ | $99.89_{\pm0.04}(0.11)$ | $100.00_{\pm0.00}(0.00)$ | $95.70_{\pm0.09}(0.01)$ | 3.19 |
| $\ell_1$-sparse MU | $100.00_{\pm0.00}(0.00)$ | $100.00_{\pm0.00}(0.00)$ | $99.99_{\pm0.01}(0.01)$ | $95.88_{\pm0.14}(0.17)$ | 2.88 |
| Class-wise forgetting, ImageNet | | | | | |
| Retrain | $100.00_{\pm0.00}$ | $100.00_{\pm0.00}$ | $71.75_{\pm0.45}$ | $69.49_{\pm0.27}$ | 26.18* |
| FT | $63.60_{\pm7.11}(36.40)$ | $68.61_{\pm9.04}(31.39)$ | $72.45_{\pm0.16}(0.70)$ | $69.80_{\pm0.23}(0.31)$ | 2.87* |
| GA | $85.10_{\pm5.92}(14.90)$ | $87.46_{\pm7.20}(12.54)$ | $65.93_{\pm0.49}(5.82)$ | $64.62_{\pm0.82}(4.87)$ | 0.01* |
| IU | $43.35_{\pm5.26}(56.65)$ | $60.83_{\pm6.17}(39.17)$ | $66.28_{\pm0.77}(5.47)$ | $66.25_{\pm0.53}(3.24)$ | 3.14* |
| $\ell_1$-sparse MU | $99.85_{\pm0.07}(0.15)$ | $100.00_{\pm0.00}(0.00)$ | $68.07_{\pm0.13}(3.68)$ | $68.01_{\pm0.21}(1.48)$ | 2.87* |
| Random data forgetting, CIFAR-10 | | | | | |
| Retrain | $5.41_{\pm0.11}$ | $13.12_{\pm0.14}$ | $100.00_{\pm0.00}$ | $94.42_{\pm0.09}$ | 42.15 |
| FT | $6.83_{\pm0.51}(1.42)$ | $14.97_{\pm0.62}(1.85)$ | $96.61_{\pm0.25}(3.39)$ | $90.13_{\pm0.26}(4.29)$ | 2.33 |
| GA | $7.54_{\pm0.29}(2.13)$ | $10.04_{\pm0.31}(3.08)$ | $93.31_{\pm0.04}(6.69)$ | $89.28_{\pm0.07}(5.14)$ | 0.31 |
| IU | $2.03_{\pm0.43}(3.38)$ | $5.07_{\pm0.74}(8.05)$ | $98.26_{\pm0.29}(1.74)$ | $91.33_{\pm0.22}(3.09)$ | 3.22 |
| $\ell_1$-sparse MU | $5.35_{\pm0.22}(0.06)$ | $12.71_{\pm0.31}(0.41)$ | $97.39_{\pm0.19}(2.61)$ | $91.26_{\pm0.20}(3.16)$ | 2.34 |
| Random data forgetting, CIFAR-100 | | | | | |
| Retrain | $24.76_{\pm0.12}$ | $49.80_{\pm0.26}$ | $99.98_{\pm0.02}$ | $74.46_{\pm0.08}$ | 48.70 |
| FT | $0.78_{\pm0.34}(23.98)$ | $1.13_{\pm0.40}(48.67)$ | $99.93_{\pm0.02}(0.05)$ | $75.14_{\pm0.09}(0.68)$ | 3.74 |
| GA | $0.04_{\pm0.02}(24.72)$ | $3.80_{\pm0.87}(46.00)$ | $99.97_{\pm0.01}(0.01)$ | $74.07_{\pm0.11}(0.39)$ | 0.24 |
| IU | $1.53_{\pm0.36}(23.23)$ | $6.58_{\pm0.42}(43.22)$ | $99.01_{\pm0.28}(0.97)$ | $71.76_{\pm0.31}(2.70)$ | 3.80 |
| $\ell_1$-sparse MU | $20.77_{\pm0.27}(3.99)$ | $36.80_{\pm0.44}(13.00)$ | $98.26_{\pm0.15}(1.72)$ | $71.52_{\pm0.21}(2.94)$ | 3.76 |
| Random data forgetting, SVHN | | | | | |
| Retrain | $4.89_{\pm0.11}$ | $15.38_{\pm0.14}$ | $100.00_{\pm0.00}$ | $95.54_{\pm0.09}$ | 42.71 |
| FT | $3.56_{\pm0.27}(1.33)$ | $10.05_{\pm0.24}(5.33)$ | $99.89_{\pm0.04}(0.11)$ | $93.55_{\pm0.12}(1.99)$ | 2.73 |
| GA | $0.99_{\pm0.42}(3.90)$ | $3.07_{\pm0.53}(12.31)$ | $99.43_{\pm0.22}(0.57)$ | $94.03_{\pm0.21}(1.51)$ | 0.26 |
| IU | $3.48_{\pm0.13}(1.41)$ | $9.44_{\pm0.27}(5.94)$ | $96.30_{\pm0.08}(3.70)$ | $91.59_{\pm0.11}(3.95)$ | 3.21 |
| $\ell_1$-sparse MU | $4.06_{\pm0.14}(0.83)$ | $11.80_{\pm0.22}(3.58)$ | $99.96_{\pm0.01}(0.04)$ | $94.98_{\pm0.03}(0.56)$ | 2.73 |

Table A7: Performance of $\ell_1$-sparse MU vs. Retrain and FT on (**Swin Transformer**, CIFAR-10).

| MU | UA | MIA-Efficacy | RA | TA | RTE (min) |
|---|---|---|---|---|---|
| Class-wise forgetting | | | | | |
| Retrain | 100.00 | 100.00 | 100.00 | 80.14 | 153.60 |
| FT | 8.56 (91.44) | 22.46 (77.54) | 99.92 (0.08) | 79.72 (0.42) | 3.87 |
| $\ell_1$-**sparse MU** | 98.80 (1.20) | 100.00 (0.00) | 98.25 (1.75) | 80.20 (0.06) | 3.89 |
| Random data forgetting | | | | | |
| Retrain | 21.48 | 28.44 | 100.00 | 78.59 | 155.06 |
| FT | 0.16 (21.32) | 1.26 (27.18) | 99.80 (0.20) | 79.54 (0.95) | 7.77 |
| $\ell_1$-**sparse MU** | 9.22 (12.26) | 18.33 (10.11) | 97.92 (2.08) | 79.09 (0.50) | 7.84 |

**Performance of $\ell_1$ sparsity-aware MU on additional architectures.** Tab. A7 presents an additional application to Swin Transformer on CIFAR-10. To facilitate a comparison between approximate unlearning methods (including the FT baseline and the proposed $\ell_1$-sparse MU) and Retrain, we train the transformer from scratch on CIFAR-10. This could potentially decrease testing accuracy compared with fine-tuning on a pre-trained model over a larger, pre-trained dataset. As we can see, our proposed $\ell_1$-sparse MU leads to a much smaller performance gap with Retrain compared to FT. In particular, class-wise forgetting exhibited a remarkable $90.24\%$ increase in UA, accompanied by a slight reduction in RA.

**Performance of 'prune first, then unlearn' and $\ell_1$ sparsity-aware MU on different model sizes.** Further, Tab. A8 and Tab. A9 present the unlearning performance versus different model sizes in the ResNet family, involving both ResNet-20s and ResNet-50 on CIFAR-10, in addition to ResNet-18 in Tab. 3. As we can see, sparsity consistently diminishes the unlearning gap with Retrain (indicated by highlighted numbers, with smaller values being preferable). It's worth noting that while both ResNet-20s and ResNet-50 benefit from sparsity, the suggested sparsity ratio is 90% for ResNet-20s and slightly lower than 95% for ResNet-50 when striking the balance between MU and generalization.

Table A8: MU performance on (**ResNet-20s**, CIFAR-10) using 'prune first, then unlearn' (applying to the OMP-resulted 90% sparse model) and 'sparse-aware unlearning' (applying to the original dense model). The performance is reported in the form $a_{\pm b}$, with mean $a$ and standard deviation $b$ computed over 10 independent trials. A performance gap against Retrain is provided in (●). Smaller performance gap from Retrain is better in the context of machine unlearning.

| MU | UA | | MIA-Efficacy | | RA | | TA | | RTE |
|---|---|---|---|---|---|---|---|---|---|
| | DENSE | 90% **Sparsity** | DENSE | 90% **Sparsity** | DENSE | 90% **Sparsity** | DENSE | 90% **Sparsity** | (min) |
| Class-wise forgetting | | | | | | | | | |
| Retrain | $100.00_{\pm 0.00}$ | $100.00_{\pm 0.00}$ | $100.00_{\pm 0.00}$ | $100.00_{\pm 0.00}$ | $99.76_{\pm 0.03}$ | $92.95_{\pm 0.20}$ | $92.22_{\pm 0.20}$ | $88.58_{\pm 0.29}$ | 25.27 |
| FT | $83.10_{\pm 4.83}$ (16.90) | $91.67_{\pm 3.81}$ (8.33) | $97.17_{\pm 0.75}$ (2.83) | $99.37_{\pm 0.29}$ (0.63) | $98.14_{\pm 0.28}$(1.62) | $93.33_{\pm 0.80}$ (0.38) | $90.99_{\pm 0.40}$ (1.23) | $88.90_{\pm 0.63}$ (0.32) | 1.57 |
| GA | $88.48_{\pm 3.47}$ (11.52) | $90.57_{\pm 2.14}$ (9.43) | $92.55_{\pm 4.27}$ (7.45) | $97.37_{\pm 2.18}$ (2.63) | $91.42_{\pm 0.53}$(8.34) | $86.75_{\pm 0.88}$ (6.20) | $85.46_{\pm 1.33}$ (6.76) | $83.33_{\pm 0.78}$ (5.25) | 0.10 |
| $\ell_1$-**sparse MU** | $98.57_{\pm 0.86}$ (1.43) | n/a | $100.00_{\pm 0.00}$ (0.00) | n/a | $96.18_{\pm 0.91}$ (3.58) | n/a | $90.18_{\pm 0.14}$ (2.04) | n/a | 1.60 |
| Random data forgetting | | | | | | | | | |
| Retrain | $8.02_{\pm 0.36}$ | $12.33_{\pm 0.38}$ | $14.94_{\pm 0.46}$ | $16.46_{\pm 0.83}$ | $100.00_{\pm 0.00}$ | $92.33_{\pm 0.18}$ | $91.10_{\pm 0.27}$ | $86.46_{\pm 0.02}$ | 25.29 |
| FT | $3.46_{\pm 0.32}$ (4.56) | $8.93_{\pm 0.52}$ (3.40) | $9.33_{\pm 0.45}$ (5.61) | $12.62_{\pm 0.51}$ (3.84) | $98.57_{\pm 0.20}$ (1.43) | $93.59_{\pm 0.33}$ (1.26) | $90.71_{\pm 0.14}$ (0.39) | $88.15_{\pm 0.12}$ (1.69) | 1.58 |
| GA | $1.84_{\pm 0.53}$ (6.18) | $6.88_{\pm 0.41}$ (5.45) | $6.53_{\pm 0.42}$ (8.41) | $9.57_{\pm 0.56}$ (6.89) | $97.41_{\pm 0.21}$ (2.59) | $94.78_{\pm 0.11}$ (2.45) | $91.03_{\pm 0.07}$ (0.07) | $89.15_{\pm 0.31}$ (2.69) | 0.10 |
| $\ell_1$-**sparse MU** | $6.44_{\pm 0.23}$ (1.58) | n/a | $13.15_{\pm 0.31}$ (1.79) | n/a | $96.31_{\pm 0.14}$ (3.69) | n/a | $89.14_{\pm 0.26}$ (1.96) | n/a | 1.58 |

Table A9: MU performance on (**ResNet-50**, CIFAR-10) following the format of Table A8.)

| MU | UA | | MIA-Efficacy | | RA | | TA | | RTE |
|---|---|---|---|---|---|---|---|---|---|
| | DENSE | 95% **Sparsity** | DENSE | 95% **Sparsity** | DENSE | 95% **Sparsity** | DENSE | 95% **Sparsity** | (min) |
| Class-wise forgetting | | | | | | | | | |
| Retrain | $100.00_{\pm 0.00}$ | $100.00_{\pm 0.00}$ | $100.00_{\pm 0.00}$ | $100.00_{\pm 0.00}$ | $100.00_{\pm 0.03}$ | $100.00_{\pm 0.00}$ | $94.18_{\pm 0.38}$ | $94.12_{\pm 0.07}$ | 96.29 |
| FT | $49.76_{\pm 5.04}$ (50.24) | $57.84_{\pm 3.10}$ (42.16) | $84.67_{\pm 6.90}$ (15.33) | $88.20_{\pm 3.70}$ (11.80) | $99.62_{\pm 0.12}$(0.38) | $99.65_{\pm 0.06}$ (0.35) | $94.11_{\pm 0.30}$ (0.07) | $93.54_{\pm 0.15}$ (0.58) | 6.02 |
| GA | $93.41_{\pm 0.24}$ (6.59) | $93.90_{\pm 0.21}$ (6.10) | $95.90_{\pm 0.18}$ (4.10) | $96.22_{\pm 0.24}$ (3.78) | $93.44_{\pm 0.53}$(6.56) | $93.05_{\pm 0.26}$ (6.95) | $87.37_{\pm 0.15}$ (6.81) | $87.22_{\pm 0.08}$ (6.90) | 0.30 |
| $\ell_1$-**sparse MU** | $96.46_{\pm 0.51}$ (3.54) | n/a | $100.00_{\pm 0.00}$ (0.00) | n/a | $99.11_{\pm 0.42}$ (0.89) | n/a | $92.83_{\pm 0.10}$ (1.35) | n/a | 6.05 |
| Random data forgetting | | | | | | | | | |
| Retrain | $5.81_{\pm 0.29}$ | $6.09_{\pm 0.45}$ | $11.99_{\pm 0.94}$ | $12.76_{\pm 0.86}$ | $100.00_{\pm 0.00}$ | $99.00_{\pm 0.00}$ | $93.62_{\pm 0.21}$ | $93.76_{\pm 0.03}$ | 96.30 |
| FT | $5.17_{\pm 0.75}$ (0.64) | $5.84_{\pm 0.45}$ (0.25) | $10.93_{\pm 0.94}$ (1.06) | $12.17_{\pm 0.82}$ (0.59) | $97.64_{\pm 0.22}$ (2.36) | $97.24_{\pm 0.35}$ (1.76) | $91.13_{\pm 0.14}$ (2.49) | $90.81_{\pm 0.12}$ (2.95) | 6.02 |
| GA | $3.42_{\pm 0.25}$ (2.39) | $5.77_{\pm 0.37}$ (0.32) | $5.20_{\pm 0.42}$ (6.79) | $8.73_{\pm 0.56}$ (4.03) | $96.20_{\pm 0.19}$ (3.80) | $95.41_{\pm 0.14}$ (3.59) | $90.12_{\pm 0.21}$ (3.50) | $89.47_{\pm 0.26}$ (4.29) | 0.32 |
| $\ell_1$-**sparse MU** | $6.13_{\pm 0.17}$ (0.32) | n/a | $12.29_{\pm 0.20}$ (0.30) | n/a | $97.12_{\pm 0.16}$ (2.88) | n/a | $91.12_{\pm 0.15}$ (2.50) | n/a | 6.10 |

# D  Broader Impacts and Limitations

**Broader impacts.** Our study on model sparsity-inspired MU provides a versatile solution to forget arbitrary data points and could give a general solution for dealing with different concerns, such as the model's privacy, efficiency, and robustness. Moreover, the applicability of our method extends beyond these aspects, with potential impacts in the following areas. ① *Regulatory compliance:* Our method enables industries, such as healthcare and finance, to adhere to regulations that require the forgetting of data after a specified period. This capability ensures compliance while preserving the utility and performance of machine learning models. ② *Fairness:* Our method could also play a crucial role in addressing fairness concerns by facilitating the unlearning of biased datasets or subsets. By removing biased information from the training data, our method contributes to mitigating bias in machine

learning models, ultimately fostering the development of fairer models. ③ *ML with adaptation and sustainability:* Our method could promote the dynamic adaptation of machine learning models by enabling the unlearning of outdated information, and thus, could enhance the accuracy and relevance of the models to the evolving trends and dynamics of the target domain. This capability fosters sustainability by ensuring that ML models remain up-to-date and adaptable, thus enabling their continued usefulness and effectiveness over time.

**Limitations.** One potential limitation of our study is the absence of provable guarantees for $\ell_1$-sparse MU. Since model sparsification is integrated with model training as a soft regularization, the lack of formal proof may raise concerns about the reliability and robustness of the approach. Furthermore, while our proposed unlearning framework is generic, its applications have mainly focused on solving computer vision tasks. As a result, its effectiveness in the domain of natural language processing (NLP) remains unverified. This consideration becomes particularly relevant when considering large language models. Therefore, further investigation is necessary for future studies to explore the applicability and performance of the framework in NLP tasks.

