# OpenReview forum: "Model Sparsity Can Simplify Machine Unlearning"
_NeurIPS.cc/2023/Conference — NeurIPS 2023 spotlight_

### Official Review · Reviewer_8xJa · 2023-06-14

**Soundness:** 3 good
**Presentation:** 4 excellent
**Contribution:** 3 good
**Rating:** 8
**Confidence:** 4

**Summary:**

This paper considers the benefit of leveraging sparsity to improve standard unlearning techniques. They empirically verify that across a wide range of datasets and architectures. the sparsity benefits unlearning.

**Strengths:**

1. The recap of this paper on the unlearning literature and metrics is very thorough and helpful for readers who are more unfamiliar with the setting of unlearning.
2. The intuition of this paper is quite elegant and helpful. The address of different pruning methodologies is insightful.
3. The empirical results are seemingly quite promising.

Overall, this paper uses a simple idea to yield strong benefits. While a theoretical analysis would have been nice, the paper is strong as it is. I vote for acceptance strongly then.

I also believe this work is missing a citation. Pruning has also been shown in in its relation to generalization error (see "Generalization Bounds for Magnitude-Based Pruning via Sparse Matrix Sketching"). This, however, does not affect my score.

**Weaknesses:**

1. There is a slight inconsistency with the empirical results. There are times when sparsity significantly hurts the unlearning. There does not seem to be an observable pattern to this. I believe such an analysis is necessary.
2. Is it possible for the authors to conduct a small experiment to see how the effect of sparsity on unlearning scales with model size? For example, with varying Resnet sizes, does the effect of sparsity on unlearning change? I think this is a slight concern for me.
3. There are no theoretical analyses here. However, I won't be too harsh about this as the paper does mention this in the limitations.

**Questions:**

1. My main question is, do the readers have an intuition for when sparsity helps or hurts the unlearning process? Across several of the experiments, while it is true that sparsity generally improves unlearning, there are several cases where this is not the case. Are there patterns the authors noticed? For a practitioner, I think a deeper understanding of this is helpful.
2. It seems that most of the architectures are restricted to convolution-based architectures. Have the authors tried extending this to other architecture, such as transformers?

**Limitations:**

I would have liked it if the authors had recognized the slight inconsistency of their empirical results. However, on the whole, the limitations are addressed.

---

> ### Author Rebuttal · Authors · 2023-08-09
>
> We greatly appreciate your insightful comments that precisely recognize the strengths of our work. We are particularly excited about citing the referenced paper that establishes a connection between model pruning and generalization. Below, we offer our detailed responses to your comments, categorized by **[W]** for weaknesses and **[Q]** for questions.
>
> **Response to W1, Q1**: Thank you for raising these questions.
>
> As previously indicated in the literature [R4, R9] and highlighted within our paper (e.g., Line 145, Tab. 3, and Fig. 3), the effectiveness of MU is assessed by its performance gap to the gold-standard retrained model (Retrain). Following this criterion, our experimental results (e.g., Tab. 2 and 3) consistently demonstrate that incorporating sparsity enhances unlearning efficacy (UA, MIT-Efficacy) and narrows the performance gap (i.e., the numbers marked by the blue color in Tab. 2 and 3) between approximate unlearning and Retrain. In **Tab. R3** presented in the attached PDF, we illustrate the averaged unlearning performance disparity against Retrain across various methods. The advantage of sparsity in unlearning is evident. On the other hand, we acknowledge that an excessively aggressive sparsity choice could potentially compromise generalization performance and/or remaining accuracy (RA). This tradeoff is akin to the one encountered in model pruning alone. Nevertheless, with an appropriate sparsity level or the integration of soft sparsity-aware regularization, we can achieve significant gains in unlearning efficacy without substantially sacrificing generalization.
>
> **Response to W2:** Thank you for your suggestion. In response, we conducted additional experiments involving both Resnet20s and Resnet50 on CIFAR-10, in addition to ResNet-18 in the paper (Tab. 3). The outcomes of these experiments are detailed in **Tab. R1** and **Tab. R2** of the attached PDF. In these new experiments, we assess the performance of both the "prune first, then unlearn" approach and the "sparsity-aware unlearning" technique (note that the latter is applied to the dense model rather than a predefined sparse model). During the rebuttal period, we opted to exclude the IU-based unlearning baseline due to its considerable computational demands and the challenge of tuning hyperparameters for optimal MU performance. Notably, our results reveal that across different model sizes, sparsity consistently diminishes the unlearning gap with Retrain (indicated by highlighted blue numbers, where smaller values are preferable). It's worth noting that while both ResNet20s and ResNet50 benefit from sparsity, the suggested sparsity ratio is 90% for ResNet20s and slightly lower than 95% for ResNet50 when striking the balance between MU and generalization.
>
> **Response to W3:** Thank you for your valuable feedback. We acknowledge that while Prop. 2 theoretically demonstrates the reduction in unlearning error of gradient ascent-based unlearning with the presence of model sparsity, this result does not universally apply to all the unlearning methods we examined, despite their promising empirical performance. To address this, we will incorporate a discussion of this limitation in the Limitations section of our paper (Appendix D).
>
> **Response to Q2:** Thank you for your insightful suggestion. We've included an additional experiment in our study, focusing on the application of Swin Transformer to CIFAR-10. This new experiment is presented in **Tab. R5** of the attached PDF. To facilitate a comparison between the assessed approximate unlearning methods (including the FT baseline and the proposed $\ell_1$-spare MU which also uses the FT loss as the unlearning objective function) and Retrain, we train the transformer from scratch on CIFAR-10. This could potentially result in a decrease in testing accuracy when compared with fine-tuning on a pre-trained model over a larger, pre-trained dataset.
>
> In Tab. R5, the results are noteworthy: in both class-wise forgetting and random data forgetting contexts, substantial enhancements were observed using our proposed $\ell_1$-spare MU, leading to a much smaller performance gap with Retrain compared to FT. In particular, class-wise forgetting exhibited a remarkable 90.24% increase in UA, accompanied by a slight reduction in RA.
>
> Motivated by this comment, we will also explore the application of our approach to language models in the future. We will incorporate this aspect into our Conclusion and Limitations sections.

---

> > ### Comment · Reviewer_8xJa · 2023-08-10
> > **Thank you to the authors**
> >
> > I acknowledge this response. I appreciate the improved experiments with larger model sizes. I maintain my score and vote to accept this paper.

---

> > > ### Author Response · Authors · 2023-08-10
> > > **Thank you!**
> > >
> > > Dear Reviewer 8xJa,
> > >
> > > We sincerely appreciate your prompt response and are pleased that you found our additional experiments beneficial. We're thrilled that your score will be maintained.
> > >
> > > Thank you once more for your valuable inputs in enhancing our submission.
> > >
> > > Best regards,

---

### Official Review · Reviewer_r13y · 2023-06-28

**Soundness:** 3 good
**Presentation:** 4 excellent
**Contribution:** 3 good
**Rating:** 7
**Confidence:** 4

**Summary:**

The authors propose to consider network sparsification as a way to improve machine unlearning (MU), the task of unlearning evidence of a given set of training samples from a neural network. In particular, they consider LTH based pruning schemes as well as a regularized training loss to complement standard MU approaches and show that this enhances the unlearning process in a variety of benchmarks and evaluation metrics. For gradient ascent-based MU, they further show an improvement in an error bound with mask-based sparsification, such as LTH pruning. Overall, this paper introduces NN sparsification to the MU domain and shows the effectiveness of this combination on standard benchmarks.

EDIT: After the fruitful discussion and additional experiments, I raise my score to 7 (Accept).

**Strengths:**

To the best of my knowledge, this is the first attempt to combine the recent advances of networks sparsification with the field of machine unlearning and hence presents itself as an original work.
The paper is clearly written and organized, which greatly eases the reading experience. The experiments are on a variety of tasks, benchmark datasets and models, which mostly support the success of combining model sparsity approaches for machine unlearning. To evaluate the successful unlearning apart from standard metrics, the authors propose a membership inference attack-based metric that allows a proper evaluation of how well a model forgot a given sample, which is an interesting and intuitive way to measure MU success.


**Weaknesses:**

The overall idea of combining sparsification with MU is not necessarily innovative, especially given that it is currently applied in almost any subfield given the hype around the LTH. While this itself is not bad---often the simplest ideas can make a lasting impact---I feel the authors make their life a bit too easy by considering all MU approaches at once and speaking about general statements that often do not hold across datasets and the breadth of presented results. Especially since the focus is on the extensive results rather than theoretical insights, I would expect a much more faceted discussion, with the different effects of sparsification on different MU approaches that is evident from the tables (see Questions). Similarly, I would expect sparsity-aware unlearning, as the suggested algorithm of this paper, to be compared to in all experiments.

**Questions:**

Given the focus of this work on the experiments (i.e., what is the impact of sparsification beyond GA approaches), my main concerns are around the results (tables are by the way very tiny), my questions are in order of appearance.

1.	The bottomline of Tab.2 drawn in the paper is that the “linearly decreasing gamma scheduler outperforms other schemes”, which I do not see, as neither approach is consistently better in all metrics, for example UA and MIA constant gamma is better. How do you explain this difference and what are the potential effects?
2.	Table 3 is broadly summarized in 5.2 as generally showing improvement of metrics through sparsification. Yet, for random data forgetting we see that the sparsification partially reduces UA and MIA, depending on the method and especially MIA is bad. For SVHN, UA and MIA completely break down. For CIFAR100, it becomes evident that we essentially have a strong trade-off between RA,TA and UA,MIA performance between original model and sparsified model. The paper would greatly benefit from insights and critical discussion of these results.
3.	One of the biggest issues I have is that sparsity-regularized unlearning does not appear in any of these experiments as comparison. This is highly suspicious, as this was introduced in this paper as another approach of model sparsification. Given the limited success of l1-based pruning in the LTH field, I would really like to see how this performs here. Moreover, the results in Fig.5 are a comparison against FT, claiming to beat FT in terms of UA and MIA. But FT was literally described as trading of UA and MIA for better RA and TA in the sparse regime before and hence is likely the easiest to beat. Please properly compare your method against all other methods on the considered benchmarks.
4.	Trojan model cleanse: Again, why is sparsity-aware unlearning not in the comparison? What would be its performance?
5.	Why do you not compare to finetuning with LTH sparsification?
6.	Proof of Prop. 2: I do not fully understand how you build the “diagonal of the mask m” after equation A12 so that diag(m)theta actually corresponds to the original mask hadprod theta. I feel this is an essential step from the known proof towards your proposition. Could you please explain in more detail?

Minor:

1.	Figure 1 is not instructive, from the visualization it is not clear what the left part ‘Data’ has to do with the rest. It is also unclear how pruning and unlearning are combined here, it is an overly simplistic prictorial of the title. I would suggest removing it to gain space for a proper discussion of the results.
2.	In proposition 1, L has not been introduced before.
3.	Sometimes you reference your methods as sparsity-regularized unlearning, sometimes sparsity-aware unlearning. Please be consistent.


**Limitations:**

As indicated by my questions, the limitations of this paper are not properly discussed in terms of the obtained results. In particular, differences in trends within a benchmark and across different datasets are not discussed, as well as the proposed regularized approach not compared to the prune-and-unlearn approach, which raises some concerns about limitations.
That being said, there is no free lunch, so a *critical* comparison and *appropriate discussion* of results and limitations answering my raised concerns would benefit the paper and consequently improve my rating.

---

> ### Author Rebuttal · Authors · 2023-08-09
>
> Thank you very much for the very insightful comment. Below, we provide our point-to-point responses to the comments, denoted by [W] for weaknesses and [Q] for questions.
>
> **W1:** The overall idea of combining sparsification with MU is not necessarily innovative.
>
> **Response to W1:** We are sorry to hear about the lack of novelty regarding our work. Please allow us to make the following clarifications.
>
> 1. As Reviewer r13y pointed out in the strengths section *"To the best of my knowledge, this is the first attempt to … hence presents itself as an original work."*, our work offers an innovative concept to probe the effect of sparsity on unlearning performance, which is substantiated by our thorough investigation across various approximate unlearning methods and metrics.
> 2. We respectfully disagree that *"our life is made a bit too easy by considering all MU approaches at once and speaking about general statements that often do not hold across datasets"*. It's important to note that we measure unlearning performance based on the smallest performance gap with the gold-standard retrained model (Retrain), as emphasized in Line 145 and indicated by blue numbers in Tab. 2 and 3. Our findings clearly demonstrate that sparsity indeed plays a pivotal role in reducing the unlearning gap across datasets (further elaborated in our response to the next question). Furthermore, our contribution goes beyond incremental advancements in the model pruning (or LTH) field. Our investigation into "Why sparsity for MU" encompasses crucial aspects:
> a. We provided theoretical insights into the benefit of model sparsity (Prop. 2).
> b. We showed that what is best for LTH (IMP) does not necessarily lead to the best unlearning performance and the pruning criteria for MU need to be carefully examined (Lines 220-257 and Fig. 4).
> c. The proposed sparsity-aware unlearning is also a great improvement in integrating sparsity with MU. The above, together with our extensive empirical studies, clearly shows the novelty of our work.
>
> **Response to Q1:** Thanks for raising this question. However, we believe this is caused by a misunderstanding of the "good" unlearning performance. As previously mentioned in the literature [R4, R9] and emphasized within our paper (e.g., Line 145, Tab. 3, and Fig. 3), the effectiveness of MU is gauged by its performance is closer to that of the gold-standard retrained model (Retrain). We invite the reviewer to revisit Tab. 2 for a detailed comparison. The decaying schedule, as evident in the table, results in the smallest performance gap with Retrain across the MU metrics highlighted in blue: (0.06, 0.41, 2.61, 3.16).
>
> **Response to Q2:** Given our prior explanation regarding Q1, we invite the reviewer to reassess the outcomes in Tab. 3. To facilitate a more accessible comparison, we extended it to **Tab. R3**. This updated table introduces a consolidated metric termed "Disparity Average," which essentially computes the average of the performance gaps between each unlearning method and Retrain across all metrics. By consulting this metric, it becomes evident that sparsity consistently yields advantages for different MU methods.
>
> **W2:** I expect sparsity-aware unlearning to be compared in all experiments.
>
> **Response to W2, Q3, Q4:** Thank you for raising these questions. We apologize for any confusion regarding the experiments on sparsity-aware unlearning.
> 1. We opted not to directly juxtapose "sparsity-aware unlearning" with "prune first, then unlearn" due to the fact that the former is employed on a dense model (unlike OMP, L1 regularization doesn't involve a hard thresholding operation on model weights), while the latter operates on a sparse model. Given the distinct initial models for MU, we were cautious about the fairness of such a comparison. Nonetheless, in response to the reviewer's query, we have extended Tab. 3 to Tab. R3 to encompass the performance outcomes of "sparsity-aware unlearning."
> 2. The reason for only comparing sparsity-aware unlearning with FT in Fig. 5 lies in the fact that the objective function used in sparsity-aware unlearning is specified by the fine-tuning (FT) loss; see Line 254. However, we did extend the scope of comparisons to include sparsity-aware unlearning vs. Retrain, FT, and IU (influence unlearning) in Tab. A6 (Appendix). We refrained from incorporating GA due to its substantial generalization drop. For a complete comparison, including GA, please refer to **Tab. R6**.
> 3. In Fig. 6, the omission of sparsity-aware unlearning in model cleansing stems from our intention to demonstrate the unlearning performance and the backdoor attack success rate for different models' sparsity levels. Since sparsity-aware unlearning is applied to a dense model (lacking the hard thresholding for model weight sparsity), it contributes just a single data point at sparsity = 0% in Fig. 6. **Fig. R1** includes the absent model cleansing performance using sparsity-aware unlearning. Clearly, it can also effectively remove the backdoor effect while largely preserving the model's generalization.
>
> **Response to Q5:** We would like to clarify that Fig. 4 has a comparison within the context of the LTH regime. Specifically, the label **Dense** signifies the FT-based model unlearning method applied directly to the dense model, while **IMP** represents the pruning approach recommended by LTH. It's worth noting that, in the figure, even though IMP exhibits enhanced generalization performance, it does not necessarily translate to improved unlearning efficacy (measured by UA and MIA-efficacy) in comparison to the 'Dense' method. In contrast, the OMP and SynFlow pruning techniques, which exhibit reduced reliance on training data, demonstrate significant improvements in model unlearning. Further details regarding this analysis can be found in Lines 220-257.
>
> For the response to the Q6 and minor questions, please refer to the **Supplement Response to Reviewer r13y** part of the General Response.

---

> > ### Comment · Reviewer_r13y · 2023-08-10
> > **Answer to rebuttal**
> >
> > Thank you for the detailed response. I acknowledge the additional work put into this rebuttal. Below, my answer to the rebuttal to make clear which points need further discussion.
> >
> > Weakness W1:
> > 1. Given the original submission, as explained in my review, I disagreed with the "thorough investigation", the comparisons seemed rather selective. I appreciate the additional information on experiments provided in the rebuttal and consider this point resolved.
> > 2. + Q1: I would appreciate an explanation on why overall performance should be close to the retrained model (rather than measured in absolute terms). What is the rationale behind similar performance serving as proxy for unlearning success?
> >
> > W2:
> > 1. I appreciate the effort. While the underlying approach is different, the resulting model is the same: a sparsified network. In theory, one could even control the sparsity of MU through \gamma (from the paper: " \gamma > 0 is a regularization parameter that controls the sparsity of the model parameters"). I don't expect the authors to fine-tune gamma to achieve a specific sparsity to squeeze into the table though, my reasoning was to compare l1-MU to retrain, i.e., how much of a gap do we see in this experiment for MU that yields some sparsity. Adding the resulting sparsity-level and adapting the discussion would be great.
> > 2. Thank you for adding this. I think especially due to that drop, it is important to have it there and critically discuss it. I want to emphasize that negative results, when properly discussed, are not bad for a paper but rather provide a full picture of the story.
> > 3. Why is it only contributing one data point? And why at 0% sparsity? Shouldn't different (non-zero) gammas give models of different (non-zero) sparsity?
> >
> > Q5: Thank you for the clarification. I have misunderstood that figure.
> >
> > I also would like to draw the authors' attention to my concerns regarding the limitations (in the Limitations section, not in the Questions or Weaknesses section), especially a lack of deeper discussion of the results. There is a plethora of experiments and advertised metrics in the paper showcasing different strength and weaknesses across methods, and between sparse and non-sparse results.

---

> > > ### Author Response · Authors · 2023-08-11
> > > **Additional Response (Part 1)**
> > >
> > > Thanks for the prompt response. We are glad to learn that our responses have resolved part of the reviewers’ questions. We also value the reviewer's insightful follow-up questions. Please find our additional responses below.
> > >
> > > **Response to [W1]:**  Thank you for your feedback. We're pleased that the extra experiments proved beneficial. In the revised version, we'll exercise caution with our wording and claims regarding a "thorough investigation." We'll also incorporate the additional experiments and their corresponding analyses/discussions.
> > >
> > > **Response to [Q1]:**  Thank you for your inquiry. The objective of machine unlearning is to establish a method for mitigating the influence of specific training data points (i.e., the forgetting dataset) on model performance. Keeping this in perspective, the "Retrain" approach involves training the model **from scratch** using the dataset that excludes the forgetting dataset. This approach serves as the benchmark, because the forgetting dataset has no impact on the retraining process. As demonstrated in [R1, R2, R3, R4, R5], Retrain is widely employed as the "ground truth" for assessing machine unlearning across diverse metrics. Furthermore, in [R1], after Def. 1 and 2, there's a discussion about the soundness of the Retrain approach. It's worth noting that while Retrain offers an "exact" performance reference but can introduce a notable computation overhead. This dilemma underscores the challenge of achieving **"approximate"** but **efficient** unlearning solutions and motivates our studies.
> > > >[R1] Thudi, Anvith, et al. Unrolling sgd: Understanding factors influencing machine unlearning. EuroS&P 2022.
> > > >
> > > >[R2] Sekhari, Ayush, et al. Remember what you want to forget: Algorithms for machine unlearning. Neurips 2021.
> > > >
> > > >[R3] Nguyen et al. A survey of machine unlearning. arXiv 2022.
> > > >
> > > >[R4] Golatkar et al. Eternal sunshine of the spotless net: Selective forgetting in deep networks. CVPR 2020.
> > > >
> > > >[R5] Xu et al. Machine Unlearning: A Survey. ACM Computing Surveys 2023.
> > > >
> > >
> > > **Response to [W2.1]:**  Thank you for highlighting this point.
> > >
> > > We apologize for any confusion caused by our earlier statement. The regularization parameter $\gamma$ indeed controls the penalty level of the $\ell_1$ norm, thereby reducing the magnitudes of "unimportant" weights. However, given a proper $\gamma$, this effect is of a "soft" nature and doesn't ensure the emergence of a "hard" sparsity pattern with **strict 0** values. Consequently, utilizing $\ell_1$ norm-based sparsity-aware unlearning isn't essential to achieve a precise weight sparsity. To justify this, we examine the weight distribution before and after employing the $\ell_1$-sparse MU used in the previous experiment Table R3. We report the obtained results below, considering class-wise forgetting (**Table R7**) and random data forgetting (**Table R8**) on (CIFAR10, ResNet-18). As we can see, the use of $\ell_1$ regularization doesn't yield the **exact 0-valued weights** (i.e., not perfect pruning), even though it does encourage more weights towards 0, as evidenced by the weight distribution comparison within the interval (1e-7, 1e-3].
> > >
> > >
> > > **Table R7: Weight distribution (class-wise forgetting)**
> > > |Weight            | Original (%)   | $\ell_1$-sparse (%) |
> > > |------------------|------------|-----------|
> > > | w=0              | 0.00       | 0.00      |
> > > | 0<w<=1e-7        | 0.00       | 3.61      |
> > > | 1e-7<w<=1e-3    | 27.86      | 66.25     |
> > > | 0.001<w<=0.05    | 71.95      | 30.02     |
> > > | 0.05<w<=0.01     | 0.14       | 0.09      |
> > > | w>0.01           | 0.05       | 0.03      |
> > >
> > > **Table R8: Weight distribution (random data forgetting)**
> > > | Range            | Original (%)   | $\ell_1$-sparse (%) |
> > > |------------------|------------|-----------|
> > > | w=0              | 0.00       | 0.00      |
> > > | 0<w<=1e-7        | 0.00       | 0.27      |
> > > | 1e-7<w<=1e-3     | 27.86      | 66.02     |
> > > | 0.001<w<=0.05    | 71.95      | 33.59     |
> > > | 0.05<w<=0.01     | 0.14       | 0.08      |
> > > | w>0.01           | 0.05       | 0.04      |
> > >
> > > Given the above observations, we wish to emphasize the distinctions between our proposed methods below.
> > >
> > > **(M1) Prune first, then unlearn:** We begin by pruning the initial dense model (e.g., using OMP). We then apply unlearning to the pruned model, preserving its "hard" sparsity pattern. The outcome is a sparse model that retains the same "hard" sparsity configuration.
> > >
> > > **(M2) $\ell_1$-sparse unlearning:** A soft regularization scheme encouraging sparsity is integrated into the unlearning objective function, which directly applies to the initial dense model. Different from “prune first, then unlearn”, the resulting model post unlearning may **not** have **strict sparsity**.
> > >
> > > Hence, in our initial submission and the first-round rebuttal, we did not directly compare M2 with M1 (that relies on a pruned model). Furthermore, our intention was to develop M2 as an extension to skip the model pruning stage needed by M1.

---

> > > > ### Author Response · Authors · 2023-08-11
> > > > **Additional Response (Part 2)**
> > > >
> > > > **(Extended analysis inspired by reviewer’s comment)**
> > > > Second, we acknowledge the reviewer's concern on M2. Thus, to enable a more effective comparison to investigate the performance of  M2 under different strict model sparsity levels, we suggest integrating a post-unlearning hard thresholding step to the resultant model from M2, which we call **$\ell_1$-sparse (strict)**. This thresholding process would effectively convert weights with small magnitudes to 0s, such as those falling within the interval (0, 1e-3] as presented in the previous **Table R7** and **R8** for M2. Given the above and following our previous experiment setup in Table R3, we present the unlearning performance of M2 against sparsity ratios in **Table R9** and **R10** below. As we can see, up to 75% sparsity, $\ell_1$-sparse MU still maintains a close performance to Retrain; see numbers in ($\cdot$) for the performance gap with Retrain under each metric. Yet, as the sparsity further increases, there is a notable drop in testing accuracy (TA). In contrast, M1 exhibits a more favorable tolerance for the sparsity ratio (95% as shown in Tab. 3), allowing for a better balance between unlearning effectiveness and retained generalization. The reason behind this difference could lie in the introduction of hard thresholding in M2 after unlearning, as opposed to the introduction of sparsity to M1 before unlearning. Consequently, the former might compromise the optimality of L1-sparse MU's outcomes due to thresholding, particularly when an excessively aggressive sparsity threshold is enforced.
> > > >
> > > > **Table R9: Results for $\ell_1$-sparse (strict) (class-wise forgetting)**
> > > > | Sparsity | UA   | MIA  | RA   | TA   |
> > > > |----------|------|------|------|------|
> > > > |Retrain | 100 | 100| 100|94.83|
> > > > | 0%      | 100(0)  | 100(0) | 98.99(1.01)| 93.40(1.43)|
> > > > | 50%     | 100(0) | 100(0)| 98.29(1.71)| 93.37(1.46)|
> > > > | 75%     | 100(0) | 100(0) | 98.15(1.85)| 92.82(2.01)|
> > > > | 90%     | 100(0) | 100(0) | 66.29(33.71)| 62.37(32.46)|
> > > > | 95%     | 100(0)| 100(0)| 12.45(87.55)| 12.46(82.37)|
> > > >
> > > > **Table R10: Results for $\ell_1$-sparse (strict) (random data forgetting)**
> > > > | Sparsity | UA   | MIA  | RA   | TA   |
> > > > |----------|------|------|------|------|
> > > > |Retrain | 5.41 | 13.12| 100|94.42|
> > > > | 0%       | 5.35(0.06)    | 12.71(0.41)   | 97.39(2.61)   | 91.26(3.16)   |
> > > > | 50%      | 5.30(0.11)     | 12.71(0.41)   | 97.39(2.61)   | 91.27(3.15)   |
> > > > | 75%      | 5.08(0.33)    | 15.51(2.39)   | 97.58(2.42)   | 91.38(3.04)   |
> > > > | 90%      | 14.33(8.92)   | 24.47(11.35)   | 88.56(11.44)   | 83.55(10.87)   |
> > > > | 95%      | 80.83(75.42)   | 53.55(40.43)   | 18.78(81.22)   | 18.44(75.98)   |
> > > >
> > > > **Response to [W2.2]:**
> > > > Thank you for the valuable feedback. We concur with the reviewer's perspective on discussing “negative results”. We are committed to enhancing our work in accordance with your suggestion.
> > > >
> > > > In addition, we wish to emphasize that the negative results (presumably referring to the observed TA drop with the GA method in **Tab. R6**) may not necessarily be attributed to the incorporation of sparsity into MU. If we examine the “TA” column and the “GA” row of Tab. 3 in the original submission, GA also induces a large TA drop for dense models in class-wise forgetting. This holds true for Tab. R6 as well. The possible reason is that GA is a simple gradient ascent method applied just to the forgetting dataset and thus, could forget the generalization ability learned from the remaining dataset. We will add this discussion in the paper. Thank you!

---

> > > > > ### Author Response · Authors · 2023-08-11
> > > > > **Additional Response (Part 3)**
> > > > >
> > > > > **Response to [W2.3]:**
> > > > > As elaborated in **W2.1**, M2 ($\ell_1$-sparse unlearning) leads to a dense model that doesn't necessarily exhibit strict sparsity. This is why it shows 0% sparsity, like the weight distribution results in **Table R7** and **R8**. However, upon including the post-unlearning hard thresholding operation to truncate weights with small magnitudes to strict zeros, we can then have the model cleanse performance of using $\ell_1$-sparse unlearning vs. model sparsity levels; see results below. We observe that the method, called $\ell_1$-sparse MU (strict), can also effectively mitigate the attack success rate. However, in contrast to the "prune first, then unlearn" approach, $\ell_1$-sparse MU (strict) might lead to a notable drop in testing accuracy (TA) as the sparsity ratio increases. This is not surprising, as we have shown in Table R9 and R10, the generalization ability of the model learned by $\ell_1$-sparse MU could be sensitive to the used hard thresholding operation for the strict sparsity level.
> > > > >
> > > > > **Table R11: TA for trojan model cleanse**
> > > > >
> > > > > | Sparsity                | 0      | 75    | 90    | 95    |
> > > > > |---------------------------|--------|-------|-------|-------|
> > > > > | $\ell_1$-sparse (strict)    | 92.76  | 91.68 | 63.79 | 26.55 |
> > > > > | Prune first then unlearn  | 93.51  | 94.06 | 93.70 | 92.82 |
> > > > >
> > > > > **Table R12: ASR for trojan model cleanse**
> > > > >
> > > > > | Sparsity                    | 0      | 75    | 90    | 95    |
> > > > > |---------------------------|--------|-------|-------|-------|
> > > > > | $\ell_1$-sparse (strict)    | 14.67  | 21.94 | 12.44 | 0.00  |
> > > > > | Prune first then unlearn  | 46.95  | 43.40 | 15.60 | 14.60 |
> > > > >
> > > > >
> > > > > **Response to Limitation:**
> > > > > Thank you for your valuable suggestions. We are committed to enhancing our discussion on the strengths and weaknesses of various unlearning methods, both sparse and non-sparse, across different datasets in our revised manuscript.
> > > > > We wholeheartedly concur with the notion that “there is no free lunch”. In light of this, we will incorporate a deeper discussion regarding the limitations of our method. This includes addressing challenges such as improving the tradeoff between unlearning efficacy and generalization, and identifying the optimal sparsity regularization schedule for $\ell_1$-sparse MU and determining the ideal sparsity level for prune first then unlearn. These complexities are further magnified when considering the variability across diverse datasets and architectures, akin to the considerations in the field of model pruning.

---

> > > > > > ### Comment · Reviewer_r13y · 2023-08-14
> > > > > > **Final thoughts and score adaptation**
> > > > > >
> > > > > > Dear authors,
> > > > > >
> > > > > > Thank you for the additional, well structured experiments and thorough explanation.
> > > > > >
> > > > > > I understand your reasoning of Q1, but still have my reservations, so please let me elaborate shortly. First, in other problem settings, such as LTH, pruned models intriguingly perform *better* than their dense counterparts as well as retrain baselines. I appreciate that this is a different problem setting (i.e., data is not reduced, but models get sparser), so let me give a short abstract argument why this might also be relevant here. Imagine a classifier tasked with detecting cows, horses, and camels. Ideally, a CNN would learn to detect individual components of each species, such as hind legs, main body, the differences in tails and head shape, fur patterns, and the camel hump, and how they are assembled together. It turned out that camels fall under privacy protection and, hence, the class along with its information should be removed. Removing from the model trained on all data ideally removes all camel-related attributes such as camel hump detectors and specific shape and fur patterns, as well as the assembly (deeper convolutions). Yet, similar to self-supervised or unsupervised pre-training, there might be shared information (e.g., leg arrangement and hove detectors) that improves detection capabilities for other classes without revealing information about the camels. This is not contained in a fully retrained model.
> > > > > > That being said, this is a more philosophical discussion that should inspire future thinking (or future work) on these problems and I won't consider my reservations for the scoring.
> > > > > >
> > > > > > The clarification with l1-sparsity training was helpful. Given the framing around sparsification and typical LASSO results, I thought the networks were "properly" sparse. I would kindly ask the authors to carefully phrase this and include the weight distributions as shown in the rebuttal.
> > > > > >
> > > > > > Trusting the authors to incorporate a proper discussion about why lunch is still not free regarding both, different sparsification methods, as well as differences in performance of sparsified vs unsparsified methods across scores (is there a systematic bias that unlearning according to some metrics is more or less difficult with sparsification), I am happy to **raise my score to Accept**. Thank you for the engaging discussion.

---

> > > > > > > ### Author Response · Authors · 2023-08-14
> > > > > > > **Thank you for your insights and raising the score**
> > > > > > >
> > > > > > > Dear Reviewer r13y,
> > > > > > >
> > > > > > > We sincerely thank you for your insightful comments with us. Yes, we agree that there exist scenarios, where the Retrain may not be a “perfect” solution to remove the influence of data points to be unlearned, especially in situations, with shared data features and spurious correlation. We concur that shared information might inadvertently aid in the texture extracting of other classes, a nuanced aspect deserving of a deeper exploration in subsequent works. This is an insightful discussion, and we will mention this limitation in the work. Moreover, we will also carefully refine our claims regarding the l1-sparse unlearning’s impact on the model sparsity.
> > > > > > >
> > > > > > > Thank you again for elevating the quality of our work and for the engaging dialogue.
> > > > > > >
> > > > > > > Best regards,

---

### Official Review · Reviewer_hHiz · 2023-07-04

**Soundness:** 3 good
**Presentation:** 2 fair
**Contribution:** 2 fair
**Rating:** 6
**Confidence:** 4

**Summary:**

This paper studies the machine unlearning problem from the perspective of model sparsification. Specifically, the paper proposes two types of model sparsification methods: data-independent (e.g., OMP) and data-dependent (e.g., sparsity-aware unlearning). An extensive set of simulations are shown in the paper to validate the unlearning performance for the sparse models.

**Strengths:**

This paper performs a comprehensive empirical study on the influence of model sparsity on the unlearned model performance, which is an intuitive but less explored direction in machine unlearning literature. The authors give some theoretical insights on how model sparsity will affect the "unlearning error" in Proposition 2, although the loss there would be restricted to be convex and the unlearning mechanism is gradient ascent. For the simulations, different types of unlearning methods are evaluated with the same pipeline to study the unlearned model performance under dense and sparse regimes. Furthermore, the authors show two possible applications of machine unlearning, namely data cleaning and transfer learning, where the model sparsity may also help with the final model performance.

**Weaknesses:**

Although this paper provides extensive simulations to show that sparse models can help increase the unlearning accuracy and the MIA efficacy while maintaining good remaining accuracy and testing accuracy, it still does not fully answer the very basic question "How to decide the level of model sparsity?" It is obvious that when we set the sparsity to 100%, the unlearned model should perform perfectly on UA and MIA-Efficacy, so there is no surprise that they should still perform well at 95% sparsity. As for the good performance on RA and TA, such observations have already been well studied in previous literature on lottery ticket hypothesis and model compression. So if one wants to incorporate sparse models for unlearning purposes, the first and most important question would be the sparsification level. Unlike previous problems where a sparse model is good when it performs well on the training statistics, in unlearning we do not know the forget set in advance, so we are not able to decide how well a sparse model will behave beforehand regarding the UA and MIA-Efficacy. The authors try to answer this question via the sparsity-aware fine-tuning, but again it falls back to decide the trade-off coefficient $\gamma$ and the current regularization scheduler is pure heuristic with little insights. Nevertheless, I understand that this is a challenging question even in general LTH problems and there would be no easy solution for that, especially in the context of unlearning.

As for the presentation of the paper, sometimes the notations or the terms are introduced without explanations. For example, in Proposition 1, the term $\mathbf{1}/N$ is used without explaining that $\theta_o=\theta(\textbf{1}/{N})$; in Proposition 2, what does the learning error $e(\mathbf{m})$ really means? Also, Proposition 1 does not seem to have any relationship with the remaining content and it is just a reformulation of the results in previous literature, so maybe it can be moved to the appendix to save space for the figures and tables.

**Questions:**

Overall this is a good empirical paper and I do not have questions about the simulations. Please see the weakness section for the concerns on methodology.

**Limitations:**

I think the biggest limitation is that it would be hard to perform theoretical analysis for the update rule Eq (3) like those influential function-based methods which require the loss to be strongly convex. Also, the current simulations are all on computer vision tasks with CNN-based models, which limits the application domain. CV models are known to be redundant and remember training samples within coefficients, so it would be good to try other tasks in different domains.

---

> ### Author Rebuttal · Authors · 2023-08-09
>
> We appreciate Reviewer hHiz for providing a detailed summary of our strengths. Below, we present our detailed responses to the comments, indicating **[W]** for weaknesses and **[Q]** for questions.
>
> **W1:** How to decide the level of model sparsity? And how to decide the trade-off coefficient $\gamma$?
>
> **Response to W1:** Thank you for posing these insightful questions. As noted by Reviewer hHiz, *"Nevertheless, I understand that determining the best sparsity level is a challenging question even in general LTH problems."* Indeed, identifying the optimal sparsity level for unlearning can be intricate, especially considering variations across datasets and architectures. For instance, similar to LTH, our empirical findings indicate that the optimal sparsity level to improve the efficacy of MU on ImageNet is approximately 80%, different from the 95% observed for CIFAR-10. While the optimal sparsity level for MU may differ, we possess some general intuition to aid in sparsity selection. The main criterion is to pinpoint a sparsity level that improves the unlearning efficacy while maintaining the generalization performance comparable to that of the original model.
>
> Furthermore, the sparsity-aware unlearning method (Eq. 3) could help circumvent the imposition of a strict threshold on model sparsity. As clarified in Lines 277-279, the optimal choice for the tradeoff coefficient $\gamma$ tends to align with a decaying scheduler, as indicated by the minimized unlearning performance gap with Retrain in Tab. 2. This schedule underscores the advantage of emphasizing sparsity enhancement during the initial stages of unlearning, gradually transitioning to heightened attention on refining fine-tuning accuracy over the retained dataset.
>
> **W2:** In Prop. 1, the term 1/N is used without explaining. In Prop. 2, the term e(m) lacks explanation.
>
> **Response to W2:** Thank you for the careful reading. $w=1/N$ signifies the uniform weights employed for Empirical Risk Minimization (**ERM**) training. Thus, $\theta(1/N)$ pertains to the original model trained via ERM, as elaborated in Line 87.
>
> Regarding $e(m)$ in Prop. 2, it pertains to the unlearning error when comparing the Gradient Ascent (**GA**)-based unlearning with the retrained model under the model sparsity mask $m$, see Eq. (A12) in Appendix B. When $m=1$ (no pruning involved), this concept was initially introduced in [Sec. 5.1, R11] to calculate the reverse GA steps that need to be reintegrated into the trained model for unlearning.
>
> **W3:** Prop. 1 is just a reformulation of the results in previous literature, moving to the appendix.
>
> **Response to W3:** We will move Prop. 1 to the Appendix. However, there's a specific reason behind the detailed exposition of IU (Prop. 1) in the initial submission. Our derived IU approach exhibits a minor yet crucial distinction from existing methods in the literature, such as [Eq. 1, R5] and [Eq. 7, R12]. As outlined in Lines 132-134, our work has accounted for the normalization effect of data influence weights ($\mathbf 1^T \mathbf w = \mathbf 1$) during the IU approach derivation. In practical terms, we have observed that IU with weight normalization outperforms existing IU methods, given their sensitivity to hyperparameter tuning.
>
> **Limitations:** The current simulations are all on computer vision tasks with CNN-based models.
>
> **Response to Limitations:** Thank you for your insightful suggestions. We've included an additional experiment in our study, focusing on the application of Swin Transformer to the CIFAR-10 dataset. This new experiment is presented in **Tab. R5** of the attached PDF. To facilitate a comparison between the assessed approximate unlearning methods (including the FT baseline and the proposed $\ell_1$-spare MU) and Retrain, we train the transformer from scratch on CIFAR-10. This could potentially result in a decrease in testing accuracy when compared with fine-tuning on a pre-trained model over a larger, pre-trained dataset.
>
> In Tab. R5, the results are noteworthy: Substantial enhancements were observed using our proposed $\ell_1$-spare MU, leading to a much smaller performance gap with Retrain compared to FT. In particular, class-wise forgetting exhibited a remarkable 90.24% increase in UA, accompanied by a slight reduction in RA.
>
> Motivated by this comment, we will also explore the application of our approach to language models in the future. We will incorporate this aspect into our Conclusion and Limitations sections.

---

> > ### Comment · Reviewer_hHiz · 2023-08-13
> > **Reviewer Response**
> >
> > I would like to thank the authors for preparing the detailed responses and additional experiments. I just have one quick follow-up question. So for the IU approach, are you trying to say that you are considering an averaged-ERM instead of the sum-ERM in previous works (i.e., [1] section 3.1)? And that makes a lot difference? If so, I am interested to learn for what metrics the new IU method outperforms existing ones.
> >
> > [1] Chuan Guo, Tom Goldstein, Awni Hannun, and Laurens Van Der Maaten, “Certified data removal from machine learning models,” arXiv preprint arXiv:1911.03030, 2019.

---

> > > ### Author Response · Authors · 2023-08-14
> > > **Additional response to Reviwer hHiz (Part 1)**
> > >
> > > Thank you for your prompt feedback. Below is our response to the follow-up question.
> > >
> > > Yes, there exist differences between IU under sum-ERM and that under ave-ERM. To provide clarity on these differences, let's repeat the notations introduced in Appendix 1.
> > >
> > > Recall that $\mathbf w$ signifies the influence weights assigned to training data points. If $w_i = 0$, then the $i$th training point $\mathbf z_i$ will be unlearned. And $L(\mathbf w,\boldsymbol\theta) =\sum_{i=1}^N [w_i L_i (\boldsymbol\theta,\mathbf z_i)]$ represents the weighted ERM loss. This loss corresponds to the ave-ERM when $\mathbf w$ is subject to the simplex constraint (i.e., $\mathbf 1^T\mathbf w=1$ and $\mathbf w\geq\mathbf 0$). By contrast, the sum-ERM does not impose the above constraint. Furthermore, let $\boldsymbol\theta_{\mathrm{o}}$ represent the original model trained through conventional ERM, which uses the weighted ERM loss with $w_i=c$ ($\forall i$) for a positive constant $c$.
> > >
> > >
> > > Given the unlearning scheme (encoded in $\mathbf w$), the IU approach aims to delineate the model parameter adjustments required by MU from the initial model $\boldsymbol\theta_{\mathrm{o}}$. Such a model weight modification is represented as$$\\Delta(\\mathbf{w}) =\\boldsymbol\\theta(\\mathbf w)-\\boldsymbol\\theta_{\\mathrm{o}},$$where $\boldsymbol\theta(\mathbf w)$ denotes the Retrain solution of using either ave-ERM or sum-ERM given $\mathbf w$, i.e., $\boldsymbol\theta(\mathbf w):=\arg\min_{\boldsymbol\theta} L(\mathbf w,\boldsymbol\theta)$ with $\boldsymbol\theta$ being optimization variables.
> > >
> > > The difference between ave-ERM and sum-ERM would play a role in  deriving $\Delta(\mathbf{w})$, since
> > > IU resorts to the first-order **Taylor expansion** of $\boldsymbol\theta (\mathbf w)$ (which is viewed as a function of $\mathbf w$).
> > >
> > > * When the sum-ERM [R1] is considered, then the linearization point is typically given by $\mathbf w = \mathbf 1$. This leads to$$\\begin{align*}\\Delta^{\\mathrm{(sum)}}(\\mathbf{w})&=\\boldsymbol\\theta(\\mathbf w)-\\boldsymbol\\theta(\\mathbf 1)\\\\&\\approx\\boldsymbol\\theta(\\mathbf 1)+\\frac{d\\boldsymbol\\theta(\\mathbf w)}{d\\mathbf w}|{\\scriptstyle\\mathbf w=\\mathbf 1}(\\mathbf w-\\mathbf 1)-\\boldsymbol\\theta(\\mathbf 1)\\\\&=\\frac{d\\boldsymbol\\theta(\\mathbf w)}{d\\mathbf w}|{\\scriptstyle\\mathbf w=\\mathbf 1}(\\mathbf w-\\mathbf 1),\\end{align*}$$where we used the fact that $\boldsymbol\theta_{\mathrm{o}}=\boldsymbol\theta(\mathbf 1)$ for sum-ERM, and
> > > $\frac{d\boldsymbol\theta(\mathbf w)}{d\mathbf w}$ is known as implicit gradient [R2] since it is defined upon an implicit optimization problem $\boldsymbol\theta(\mathbf w)=\arg\min_{\boldsymbol\theta} L(\mathbf w,\boldsymbol\theta)$.
> > >
> > > * When the ave-ERM (Appendix 1) is considered, the linearization point is given by $\mathbf w=\mathbf 1/N$. This leads to$$\\begin{align*}\\Delta^{\\mathrm{(ave)}}(\\mathbf{w})&=\\boldsymbol\\theta(\\mathbf w)-\\boldsymbol\\theta(\\mathbf 1/N)\\\\&\\approx\\boldsymbol\\theta(\\mathbf 1/N)+\\frac{d\\boldsymbol\\theta(\\mathbf w)}{d\\mathbf w} |{\\scriptstyle\\mathbf w=\\mathbf 1/N}(\\mathbf w-\\mathbf 1/N)-\\boldsymbol\\theta(\\mathbf 1/N)\\\\&=\\frac{d\\boldsymbol\\theta(\\mathbf w)}{d\\mathbf w}|{\\scriptstyle\\mathbf w=\\mathbf 1/N}(\\mathbf w-\\mathbf 1/N),\\end{align*}$$where we used the fact that $\boldsymbol\theta_{\mathrm{o}}=\boldsymbol{\theta}(\mathbf 1/N)$ for ave-ERM.
> > > Note that the derivation of the implicit gradient $\frac{d\boldsymbol\theta(\mathbf w)}{d\mathbf w}$ has been provided in Appendix 1.

---

> > > > ### Author Response · Authors · 2023-08-14
> > > > **Additional response to Reviwer hHiz (Part 2)**
> > > >
> > > > If we compare $\Delta^{\mathrm{(sum)}}(\mathbf{w})$ with $\Delta^{\mathrm{(ave)}}(\mathbf{w})$, there exist two differences.
> > > >
> > > > 1. **$(\mathbf w - \mathbf 1)$ in sum-ERM vs. $(\mathbf w-\mathbf 1/N)$ in ave-ERM:**
> > > >
> > > >    For example, if we aim to unlearn the first $k$ training data points, the unlearning weights $\mathbf w_{\mathrm{MU}}$ under sum-ERM is then given by  $\mathbf w_{\mathrm{MU}}^{\mathrm{(sum)}} = [\underbrace{0, 0, \ldots, 0}_{k~ 0s}, 1, 1,\ldots, 1],$
> > > >
> > > >    where $0$ encodes the data sample to be unlearned or removed. This yields $(\mathbf w_{\mathrm{MU}}^{\mathrm{(sum)}} - \mathbf 1) = [\underbrace{1, 1, \ldots, 1}_{k ~ 1s}, 0, 0, \ldots, 0]$.
> > > >
> > > >    By contrast, the unlearning weights $\mathbf w_{\mathrm{MU}}$ under ave-ERM is then given by  $\mathbf w_{\mathrm{MU}}^{\mathrm{(ave)}} = [\underbrace{0, 0, \ldots, 0}_{k~ 0s}, \frac{1}{N-k}, \frac{1}{N-k},\ldots, \frac{1}{N-k}].$
> > > >
> > > >    As a result, $(\mathbf w_{\mathrm{MU}}^{\mathrm{(ave)}}-\mathbf 1/N) = [\underbrace{-\frac{1}{N}, -\frac{1}{N}, \ldots, -\frac{1}{N}}_{k~ \frac{1}{N}s}, \frac{1}{N-k} - \frac{1}{N}, \frac{1}{N-k}- \frac{1}{N},\ldots, \frac{1}{N-k}- \frac{1}{N}].$
> > > >
> > > >    The above difference is caused by the presence of simplex constraint of $\mathbf w$ in ave-ERM. Thus, the MU's weight configuration $(\mathbf w_{\mathrm{MU}}^{\mathrm{(ave)}}-\mathbf 1/N)$ obtained from ave-ERM is different from $(\mathbf w_{\mathrm{MU}}^{\mathrm{(sum)}} - \mathbf 1)$ in the sum-ERM setting.
> > > >
> > > > 2. **The implicit gradient (IG) $\frac{d\boldsymbol\theta(\mathbf w)}{d\mathbf w}|{\scriptstyle\mathbf w= \mathbf 1}$ in sum-ERM vs. $\frac{d\boldsymbol\theta(\mathbf w)}{d\mathbf w}|{\scriptstyle\mathbf w= \mathbf 1/N}$ in ave-ERM.**
> > > >
> > > >    The IG is also different as it is evaluated at two different linearization points. This will affect the calculations of IU.
> > > >
> > > > In summary, the accuracy of the Taylor expansion might be affected by using ave-ERM vs. sum-ERM. Given the example mentioned in the above Point 1, the error term of the Taylor expansion using sum-ERM for $\mathbf w = \mathbf w_{\mathrm{MU}}^{\mathrm{sum}}$ is in the order of $\|\mathbf w_{\mathrm{MU}}^{\mathrm{sum}} - \mathbf 1\|_2^2=k,$
> > > >
> > > > while the error term using ave-ERM for $\mathbf w = \mathbf w_{\mathrm{MU}}^{\mathrm{ave}}$ is in the order of$\|\mathbf w_{\mathrm{MU}}^{\mathrm{ave}} - \mathbf 1/N\|_2^2=\frac{k}{N^2} + \frac{k^2}{N^2(N-k)} = \frac{k}{N(N-k)}.$Thus compared to ave-ERM, the use of sum-ERM could cause the first-order Taylor expansion in IU less accurate as the number of unlearning data points ($k$) increases.
> > > >
> > > > We hope the above response has answered your question properly.
> > > >
> > > > > [R1] Chuan Guo, Tom Goldstein, Awni Hannun, and Laurens Van Der Maaten, “Certified data removal from machine learning models,” arXiv preprint arXiv:1911.03030, 2019.
> > > > >
> > > > > [R2] Gould, Stephen, et al. "On differentiating parameterized argmin and argmax problems with application to bi-level optimization." arXiv preprint arXiv:1607.05447 (2016).

---

> > > > > ### Comment · Reviewer_hHiz · 2023-08-14
> > > > >
> > > > > Thank you for the detailed explanations. I do not have further questions.

---

> > > > > > ### Author Response · Authors · 2023-08-14
> > > > > > **Thank you!**
> > > > > >
> > > > > > Dear Reviewer hHiz,
> > > > > >
> > > > > > Thank you once again for your diligent review and valuable feedback. We greatly appreciate your acknowledgment of our efforts to address your concerns. Your insights have been instrumental in improving our work, and we are committed to incorporating your suggestions into the revision. We would also greatly appreciate your consideration in possibly raising the original rating (6) if you find our responses satisfactory. However, if you believe there are any remaining areas where additional clarifications/responses could contribute to such a higher rating, please don't hesitate to inform us. Your guidance is always crucial to improve the quality of our submission.
> > > > > >
> > > > > > Thank you very much,
> > > > > >
> > > > > > Authors

---

### Official Review · Reviewer_MsfJ · 2023-07-07

**Soundness:** 4 excellent
**Presentation:** 4 excellent
**Contribution:** 3 good
**Rating:** 7
**Confidence:** 4

**Summary:**

The paper proposes that model sparsity leads to models that are easier to "unlearn from". The authors discuss in depth the technical measures that are and that should be used to evaluate various methods of unlearning, and suggest and demonstrate that sparsity is an effective tool in boosting these measures across a wide variety of unlearning applications.

**Strengths:**

The paper strongly supports its main claim, with extensive discussion and experimental evaluation that shows beyond much doubt that model sparsity leads to easier and stronger unlearning results in practice.

The writing is extremely clear and all internal referencing and definitions help readers easily follow and understand the motivations and results. I particularly appreciate the use of emphasis and acronyms, and the way in which the paper makes it easy to flip back and forth to find where a term was defined or where a term is used.

Weaknesses and questions below notwithstanding, the paper is quite solid albeit with a narrow scope. I appreciate the time and detail that went into the experimental evaluations.


**Weaknesses:**

I have two main concerns:

1) The authors don't seem to discuss or acknowledge that sparsity seems to lead to simply better (read: better generalizing) models, and as such we would expect better generalization to lead to less dependence on specific subsets of the training data, and thus easier unlearning. In essence, how does sparsity as a proxy for generalization aid unlearning more than other generalization methods? I would of course expect that strong regularization would affect performance, but perhaps other methods for better generalization that maintain performance may also "simplify machine unlearning".

2) My other main concern with this work is the sidelining of the body of work on $\epsilon-\delta$ forgetting. The authors reference this work in their Related Work section (probabilistic, DP), but there are many places within the main text up to that point where I was left wondering how those approaches may stack up. Simple methods such as Guo et al.'s [54] seem like an easy enough place to do a quick comparison. I acknowledge that those methods tend to depend heavily on hyperparameter choices (as stated by the authors), but a lot of that work seems highly relevant. The discussion in Section 2 around Proposition 1 is very suggestive and brings to mind work in Sekhari [57], and this citing paper seems to be trying to solve a similar problem as the authors here with approximating the Hessian inverse:
Deep Unlearning via Randomized Conditionally Independent Hessians. Ronak Mehta, Sourav Pal, Vikas Singh, Sathya N. Ravi. CVPR 2022.
(not major: I wonder if updating a subset of parameters is "similar" in some form to a sparsity approach?)

**Questions:**

1. Why were approximate methods in eps-delta not compared? If the authors strongly feel that this is out of scope I think it needs to be adequately justified.
2. Classwise and random-data methods were primarily evaluated; why not individual samples?
3. Gradient norms are often used to evaluate removal success, any reason why there were excluded?
4. Some main paper experimental results were limited to 95% sparsity, and there are a large number of references to the appendix with additional results.

Minor:
1. Depending on how the authors treat the approximate/DP/eps-delta, those might be included in the "approximate MU methods" in Section 2; I was concerned that this highly relevant work was not mentioned as I was reading through.
2. The metrics described in Section 2 are very similar to the "read-out functions" in [12], might be worth mentioning/referencing.
3. It could be helpful to clearly indicate "higher is better" "lower is better" for the various metrics, perhaps using simple up or down arrows, in the text and and in the tables. I do appreciate the authors detailed discussion of results however, careful reading covers all bases. Just thought it may help people skimming.

**Limitations:**

1. It's unclear if sparsity-promoting methods help unlearning moreso than other methods that improve model performance generally.
2. A reasonable set of related work in the form of $(\epsilon,\delta)$ forgetting is largely left un-evaluated.

---

> ### Author Rebuttal · Authors · 2023-08-09
>
> We thank Reviewer MsfJ for acknowledging the contributions, soundness, and presentation quality of our paper. And greatly appreciate Reviewer MsfJ for proposing these insightful questions. Below, we provide our responses to the comments, denoted by **[W]** for weaknesses and **[Q]** for questions.
>
> **Response to W1:**
> As shown in L162-165 and Fig. 2, sparsity indeed benefits model generalization, particularly when using iterative magnitude pruning (**IMP**). However, we refrain from concluding that better generalization simplifies and improves machine unlearning (**MU**), as the method of achieving generalization improvement strongly influences MU performance. For instance, in the comparison of pruning methods shown in Fig. 4, IMP exhibits the best generalization performance compared to other pruning methods (OMP and SynFlow) but leads to the worst unlearning accuracy (with the largest gap from Retrain). The reason is IMP's strong dependence on the forgetting dataset, as stated in L224-225 and L252-255.
>
> Thus, generalization improvement alone may not be a precise indicator for easier unlearning; it depends on the approach used to achieve generalization. To further substantiate this point, we performed additional experiments using Sharpness-aware minimization (**SAM**) [R1] during model training to enhance generalization before unlearning; see **Tab. R4** in the attached PDF. We did not observe a significantly reduced performance gap between FT and Retrain (against the former's variance) when compared to empirical risk minimization (**ERM**) training.
>
> We propose that *if the generalization improvement method does not rely on additional dependence on the forgetting dataset, it could serve as an indicator for easier unlearning*. In such cases, improved generalization may suggest that the model suffers less from spurious correlations [R2] in the training data, potentially aiding in unlearning, as suggested by the reviewer. However, a more comprehensive investigation is warranted. This question poses valuable insights for future research.
>
> **Response to W2 & Q1:** First, we will include a discussion on $\epsilon-\delta$ forgetting in Sec. 2. We will highlight its connection with influence unlearning (**IU**). Recent unlearning works [Sec 2.2, R3], [Sec 5.1, R4] have considered $\epsilon-\delta$ forgetting as part of IU.
>
> Second, our focus in this paper is on efficient approximate unlearning on pre-trained models. However, the MU approach in [R5] requires modifying the model training pipeline and integrating it into the certified data removal process (Algorithm 1, 2 of [R5]). In addition, their MU paradigm is limited to linear classifiers or linear probing, which only updates the linear classification head for DL models. This setup differs from ours, where we investigate unlearning on the full DL model. This limitation was also noted in [Tab. 4, R6; Related work, R7]. Upon reviewing the implementation code of [R5], we found that even in the case of linear probing, they considered binary classifiers, rather than the prediction head ResNet used. Considering these factors, [R5] may not be an ideal candidate for efficient unlearning comparison, although we are happy to provide further elaboration on the distinctions in the paper.
>
> Third, we sincerely appreciate your suggestion to consider reference [R8]. It indeed provides relevant insights into MU. In that work, they utilized a portion of the parameters to approximate the inversion of the Hessian matrix, enhancing the Hessian-based (IU) method. In contrast, our study focuses on revealing a crucial factor, weight sparsity, which impacts various MU methods. Our research encompasses both practical and theoretical aspects (Sec. 3 & Sec. 5), novel MU methods (Sec. 4), and emerging MU applications (Sec. 5.2).
>
> **Response to Q2:** We chose not to consider individual samples for unlearning due to several compelling reasons.
>
> First, we did not explore unlearning individual samples, as it can lead to substantial variance in unlearning performance depending on the selected forgetting sample. For instance, the UA for an individual sample would be either 100 or 0, resulting in significant variability that hampers meaningful comparisons across different settings or methods.
>
> Additionally, we performed a literature review to validate the prevalence of class-wise and random data forgetting as primary unlearning settings. Supporting evidence for our approach can be found in [Tab. 1, R9], [Fig. 2, R7] for class-wise forgetting, and [Tab. 1, R9] along with [Sec 5.1, R10] for random data forgetting.
>
> Lastly, we focus on random data forgetting and class-wise forgetting due to their direct relevance to the applications discussed in our paper. The former is aligned with the use case of model cleansing, while the latter is particularly relevant for enhancing transfer learning performance.
>
> **Response to Q3**: We admit that *gradient residual norms* (**GRN**) could be a useful metric for evaluating MU methods. Although this was introduced by [R5] to offer insights into approximation errors, we exclude it due to the following reason.
>
> Existing unlearning metrics center around three primary aspects: 1) efficiency (RTE); 2) fidelity (RA, TA); 3) efficacy (UA, MIA-Efficacy). In relation to GRN, it unveils the convergence of model retraining over the retained dataset, making it akin to a fidelity metric similar to RA (remaining accuracy). As evidenced in [Fig. 5, R8], GRN exhibits a closely aligned trend with RA under the same architecture, albeit with greater variance. Hence, we opt for RA to provide a more intuitive performance measure.
>
> **Response to Q4:** We will move more results from the appendix to the main paper in our revised version.
>
> For the response to the minor questions, please refer to the **Supplement Response to Reviewer MsfJ** part of the General Response.

---

> > ### Comment · Reviewer_MsfJ · 2023-08-14
> >
> > I acknowledge the authors' responses to my questions and others. I am quite satisfied with the responses and have increased my score.
> >
> > I do have some concern about how much work has been done during the review/rebuttal period, but I don't think that takes away from the authors' quality submission. Unfortunately that's how the game is played now...

---

> > > ### Author Response · Authors · 2023-08-14
> > > **Thank you!**
> > >
> > > Dear Reviewer MsfJ,
> > >
> > > Thank you for your swift response and for recognizing our efforts in addressing your previous questions. We are pleased to hear that our responses have been satisfactory. We will certainly make the revisions as discussed to enhance the quality of our work.
> > >
> > > Thanks,
> > >
> > > Authors,

---

### Author Rebuttal · Authors · 2023-08-09

Dear Reviewers, ACs, and PCs:

We are glad to receive valuable and constructive comments from all the reviewers. We have made a substantial effort to clarify reviewers' doubts and enrich our experiments in the rebuttal phase. In our responses, **Tab. R**xx or **Fig. R**xx refers to the new **R**ebuttal results in the attached PDF. And **Tab. A**xx or **Fig. A**xx refers to the existing results in **A**ppendix. Below is a summary of our responses:

**Reviewer [MsfJ](https://openreview.net/forum?id=0jZH883i34&noteId=JyAsNNFA2c):**
1. We endeavored to elucidate the relationship between model generalization and unlearning efficacy, supplemented by additional experiments (**Tab. R4**) to substantiate our viewpoint.
2. We clarified the importance of ε - δ forgetting and the rationale behind not using it as a comparison method in our work.
3. We provided an explanation for our decision not to use individual samples as an unlearning setting.
4. We provided clarification regarding our decision not to use the gradient residual norm as an evaluation metric.

**Supplement Response to Reviewer [MsfJ](https://openreview.net/forum?id=0jZH883i34&noteId=JyAsNNFA2c):**

- **Response to Minor Q1:** Thank you for bringing this to our attention. We will enhance Sec. 2 by offering a more comprehensive discussion about the methodologies of approximate/DP/$\epsilon-\delta$. Furthermore, we will explicitly elucidate why these methods were not incorporated in our study, providing clear explanations that align with the responses we have already given for W2 and Q1.
- **Response to Minor Q2:** Thank you for pointing this out. We will mention this in revision.
- **Response to Minor Q3:** Thank you for bringing this to our attention. As highlighted in Lines 144-146, an unlearned model exhibiting performance closer to Retrain should be deemed superior. This explains our practice of presenting the performance gap with Retrain in blue within our results (e.g., Tab. 2 and Tab. 3). Consequently, drawing a simple conclusion that higher values are always superior, or vice versa, might not accurately capture the scenario.

**Reviewer [hHiz](https://openreview.net/forum?id=0jZH883i34&noteId=0l9gILoXgT):**
1. We have provided further clarification on how we determine the pruning ratio and the parameters of sparsity-aware unlearning.
2. We included an experiment on MU using Swin Transformer (**Tab. R5**).

**Reviewer [r13y](https://openreview.net/forum?id=0jZH883i34&noteId=QsW2VshsPd):**
1. We provided extra clarifications regarding our contributions.
2. We further clarified the evaluation metrics and improved the presentation of the main table (**Tab. R3**).
3. We extended the main tables to include a more thorough comparison of the unlearning performance between the sparsity-aware unlearning and the "prune first, then unlearn" approach (**Tab. R3 and R6**).
4. We included sparsity-aware unlearning on the Trojan model cleanse application (**Fig. R1**).

**Supplement Response to Reviewer [r13y](https://openreview.net/forum?id=0jZH883i34&noteId=QsW2VshsPd):**

- **Response to Q6:** Thank you for pointing out this. We used the following facts: $diag(m) = [ m_1, 0, …, 0; 0, m_2, 0, …, 0; …; 0, …, 0, m_d ]$ is a diagonal matrix with the diagonal line given by the d-dimensional vector $m$, and thus the matrix-vector product yields $diag(\mathbf m)\boldsymbol \theta = \mathbf m \odot \boldsymbol \theta $, where $\odot$ is the element-wise product. We will make this clearer in our revision.
- **Response to Minor Questions:** We appreciate your suggestion regarding the visual representation in Fig. 1. We will certainly consider your recommendation to remove the teaser figure and provide a more detailed results analysis. Regarding Prop. 1, "L" corresponds to the empirical risk minimization loss function, as indicated in Line 123. Additionally, we will make sure that our references to "sparsity-aware unlearning" remain consistent throughout the paper.

**Reviewer [8xJa](https://openreview.net/forum?id=0jZH883i34&noteId=pIntq1EXoH):**
1. We provided explanations on the observed inconsistency in our results and revised the main table (**Tab. R3**).
2. We included additional experiments on the ResNet-20 and ResNet-50 architectures (**Tab. R1, R2**).
3. We conducted an additional experiment on the SwinTransformer architecture (**Tab. R5**).

**References used in authors' response:**

> [R1] Foret et al. Sharpness-aware minimization for efficiently improving generalization. ICLR 2021.
>
> [R2] Sagawa et al. An investigation of why overparameterization exacerbates spurious correlations. ICML 2020.
>
> [R3] Wang et al. Federated unlearning via class-discriminative pruning. WWW 2022.
>
> [R4] Xu et al. Machine Unlearning: A Survey. ACM Computing Surveys 2023.
>
> [R5] Guo et al. Certified data removal from machine learning models. ICML 2020.
>
> [R6] Nguyen et al. A survey of machine unlearning. arXiv 2022.
>
> [R7] Graves et al. Amnesiac machine learning. AAAI 2021.
>
> [R8] Mehta et al. Deep unlearning via randomized conditionally independent hessians. CVPR 2022.
>
> [R9] Golatkar et al. Eternal sunshine of the spotless net: Selective forgetting in deep networks. CVPR 2020.
>
> [R10] Golatkar et al. Forgetting outside the box: Scrubbing deep networks of information accessible from input-output observations. ECCV 2020
>
> [R11] Thudi et al. Unrolling sgd: Understanding factors influencing machine unlearning. EuroS&P 2022.
>
> [R12] Warnecke et al. Machine unlearning of features and labels. NDSS 2021.

---

### Decision · Program_Chairs · 2023-09-21

**Decision:**

Accept (spotlight)

**Comment:**

I would like to congratulate both the authors and the reviewers for a very productive round of reviews, rebuttals, comments and replies. The paper went initially from an average < 6 to an average of 7.

I personally like the global message carried by the paper.

The authors now have to take the necessary steps to improve the submission, including the experiments, technical content, up to the precision level asked for by reviewers (e.g. r13y).